# UltraLIF: Fully Differentiable Spiking Neural Networks via Ultradiscretization and Max-Plus Algebra

**Jose Marie Antonio Miñoza** [1]

## Abstract

Spiking Neural Networks (SNNs) offer energy-efficient, biologically plausible computation but suffer from non-differentiable spike generation, necessitating reliance on heuristic surrogate gradients. This paper introduces **UltraLIF**, a principled framework that replaces surrogate gradients with *ultradiscretization*, a mathematical formalism from tropical geometry providing continuous relaxations of discrete dynamics. The central insight is that the max-plus semiring underlying ultradiscretization naturally models neural threshold dynamics: the log-sum-exp function serves as a differentiable soft-maximum that converges to hard thresholding as a learnable temperature parameter $\varepsilon \to 0$. Two neuron models are derived from distinct dynamical systems: UltraLIF from the LIF ordinary differential equation (temporal dynamics) and UltraDLIF from the diffusion equation modeling gap junction coupling across neuronal populations (spatial dynamics). Both yield fully differentiable SNNs trainable via standard backpropagation with no forward-backward mismatch. Theoretical analysis establishes pointwise convergence to tropical LIF dynamics with quantitative error bounds and bounded non-vanishing gradients. Experiments on six benchmarks spanning static images, neuromorphic vision, and audio demonstrate improvements over surrogate gradient baselines, with gains most pronounced in the ultra-low latency regime ($T{=}1$) on neuromorphic and temporal datasets. An optional sparsity penalty enables significant energy reduction while maintaining competitive accuracy.

[1]Center for AI Research PH. Correspondence to: Jose Marie Antonio Miñoza <jminoza@upd.edu.ph>.

*Proceedings of the 43rd International Conference on Machine Learning*, Seoul, South Korea. PMLR 306, 2026. Copyright 2026 by the author(s).

## 1. Introduction

Spiking Neural Networks (SNNs) represent a promising paradigm for energy-efficient machine learning, with significant potential for neuromorphic hardware deployment (Maass, 1997; Roy et al., 2019). In contrast to artificial neural networks (ANNs) communicating via continuous activations, SNNs process information through discrete spike events, emulating biological neural computation. This event-driven nature enables substantial energy savings; Intel's Loihi chip demonstrates up to $1000\times$ energy reduction compared to GPUs on certain tasks (Davies et al., 2018).

However, SNN training remains challenging due to the *non-differentiability* of spike generation. The standard Leaky Integrate-and-Fire (LIF) neuron follows the dynamics:

$$v^{(t+1)} = \tau_0 v^{(t)} + I^{(t)} \tag{1}$$
$$s^{(t+1)} = H(v^{(t+1)} - \theta) \tag{2}$$

where the Heaviside step function $H(\cdot)$ has gradient zero almost everywhere. The dominant approach employs *surrogate gradients*, replacing the true gradient with a smooth approximation during backpropagation (Neftci et al., 2019; Zenke & Vogels, 2021). While empirically effective, surrogate gradients introduce a fundamental mismatch between forward (discrete) and backward (continuous) passes (Figure 1a), with limited theoretical understanding of convergence properties (Li et al., 2021; Gygax & Zenke, 2025).

This paper proposes **UltraLIF**, a theoretically grounded alternative based on *ultradiscretization*, a limiting procedure from tropical geometry transforming continuous dynamical systems into discrete max-plus systems while preserving structural properties (Tokihiro et al., 1996; Grammaticos et al., 2004). The key contributions are:

1. **Principled differentiability**: The LSE function provides a natural soft relaxation of the max operation underlying spike generation, with explicit convergence bounds as $\varepsilon \to 0$ (Lemma 3.2).
2. **Forward-backward consistency**: Unlike surrogate methods, UltraLIF employs identical dynamics in forward and backward passes, eliminating gradient mismatch (Remark 5.8).
3. **Bounded gradients**: For any $\varepsilon > 0$, gradients remain

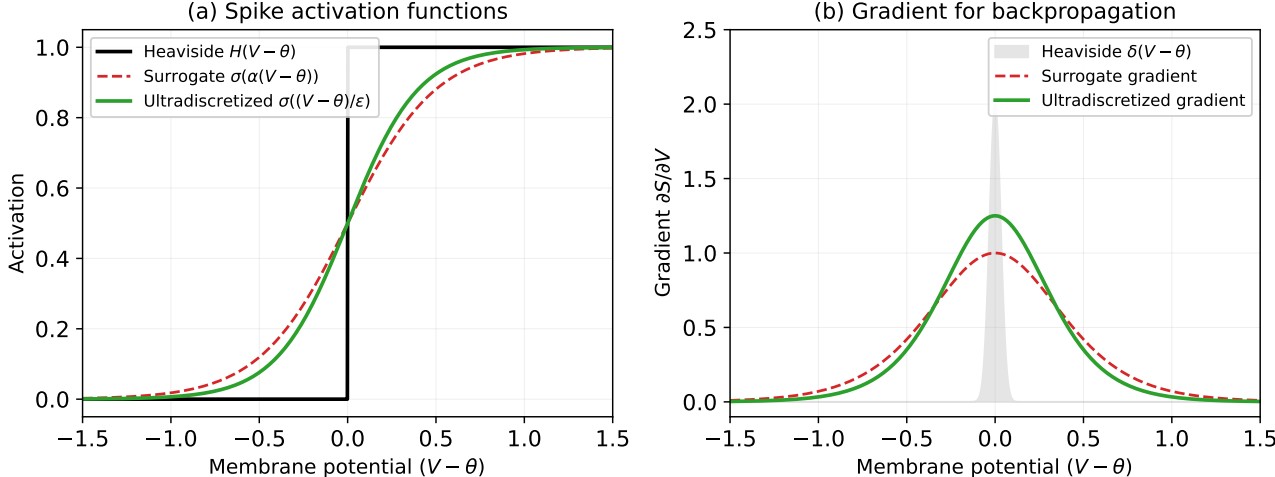

*Figure 1.* **(a)** Spike activation functions: Heaviside (hard threshold), surrogate gradient (smooth approximation), and ultradiscretized (principled soft relaxation). **(b)** Gradients: Heaviside has zero gradient almost everywhere (delta function at threshold); surrogate and ultradiscretized provide smooth gradients, but only ultradiscretized maintains forward-backward consistency.

bounded and non-vanishing, enabling stable optimization (Proposition 5.4).

4. **Ultra-low latency performance**: On six benchmarks spanning static, neuromorphic, and audio modalities, ultradiscretized models achieve state-of-the-art in the ultra-low latency regime ($T{=}1$), with the largest gains on temporal and event-driven data (+11.22% SHD, +7.96% DVS-Gesture, +3.91% N-MNIST).

**Conflict of Interest Disclosure.** The author declares no financial conflicts of interest.

## 2. Related Work

**Surrogate Gradient Methods.** The dominant paradigm for direct SNN training replaces non-differentiable spike gradients with smooth surrogates (Neftci et al., 2019). Common choices include piecewise linear (Bellec et al., 2018), sigmoid (Zenke & Ganguli, 2018), and arctangent (Fang et al., 2021b) functions. Recent work introduces learnable surrogate parameters (Lian et al., 2023) and adaptive shapes (Li et al., 2021). Yao et al. (2022) unify membrane dynamics, threshold, and reset into a generalized LIF (GLIF) framework with jointly learned parameters. Despite empirical success, the forward-backward mismatch remains theoretically problematic. Gygax & Zenke (2025) provide partial justification via stochastic neurons, showing surrogate gradients match escape noise derivatives in expectation.

**Spike Timing Approaches.** SpikeProp (Bohte et al., 2002) and variants (Mostafa, 2018; Kheradpisheh & Masquelier, 2020) compute exact gradients with respect to spike times. These methods require careful initialization and struggle with silent neurons. Recent work on exact smooth gradients

through spike timing (Göltz et al., 2021) addresses some limitations but remains computationally intensive.

**ANN-to-SNN Conversion.** An alternative approach trains conventional ANNs then converts to SNNs (Cao et al., 2015; Rueckauer et al., 2017; Bu et al., 2022). While avoiding direct SNN training, conversion methods typically require many timesteps to achieve ANN accuracy and sacrifice temporal dynamics. Dedicated ultra-low latency training (Chowdhury et al., 2021) reduces inference to $T{=}1$ via iterative timestep compression, achieving 93.05% on CIFAR-10 with VGG16 using surrogate gradients; UltraLIF reaches 93.37% with ResNet18 via principled ultradiscretization, and additionally handles neuromorphic datasets where $T{=}1$ matters most.

**Tropical Geometry and Neural Networks.** Tropical geometry studies algebraic structures where addition becomes max (or min) and multiplication becomes addition (Maclagan & Sturmfels, 2015). Zhang et al. (2018) establish that ReLU networks compute tropical rational functions. Recent work extends this to graph neural networks (Pham & Garg, 2024) and neural network compression (Fotopoulos et al., 2024). The connection to spiking networks via ultradiscretization appears novel.

**Ultradiscretization.** Originating in integrable systems (Tokihiro et al., 1996), ultradiscretization transforms difference equations into cellular automata preserving solution structure. Applications include soliton systems (Takahashi & Satsuma, 1990) and limit cycle analysis (Yamazaki & Ohmori, 2023; 2024). Application to neural networks has not been previously explored.

## 3. Preliminaries

### 3.1. The Max-Plus Semiring

**Definition 3.1** (Max-Plus Semiring). The *max-plus semiring* is the algebraic structure $(\mathbb{R} \cup \{-\infty\}, \oplus, \odot)$ where:

- $a \oplus b := \max(a, b)$ (tropical addition)

- $a \odot b := a + b$ (tropical multiplication)

with additive identity $-\infty$ and multiplicative identity $0$.

This semiring underlies tropical geometry and provides the limit structure for ultradiscretization.

### 3.2. Log-Sum-Exp as Soft Maximum

The log-sum-exp function with temperature $\varepsilon > 0$ is defined as:

$$\mathrm{LSE}_\varepsilon(\mathbf{x}) := \varepsilon \log \left( \sum_{i=1}^n e^{x_i/\varepsilon} \right) \tag{3}$$

The following lemma establishes its role as a smooth approximation to the maximum.

**Lemma 3.2** (LSE Convergence). *Let* $\mathbf{x} = (x_1, \ldots, x_n) \in \mathbb{R}^n$ *and* $M := \max_i x_i$. *Then* $M \leq \mathrm{LSE}_\varepsilon(\mathbf{x}) \leq M + \varepsilon \log n$, *hence* $\lim_{\varepsilon \to 0^+} \mathrm{LSE}_\varepsilon(\mathbf{x}) = M$. *Moreover,* $\nabla \mathrm{LSE}_\varepsilon(\mathbf{x}) = \mathrm{softmax}(\mathbf{x}/\varepsilon) \in (0, 1)^n$.

*Proof.* For the lower bound, the sum includes $e^{M/\varepsilon}$, so $\mathrm{LSE}_\varepsilon(\mathbf{x}) \geq \varepsilon \log(e^{M/\varepsilon}) = M$. For the upper bound, since $x_i \leq M$ for all $i$, $\mathrm{LSE}_\varepsilon(\mathbf{x}) \leq \varepsilon \log(n \cdot e^{M/\varepsilon}) = M + \varepsilon \log n$. The limit follows by the squeeze theorem. The gradient formula follows from direct differentiation: $\frac{\partial \mathrm{LSE}_\varepsilon}{\partial x_j} = \frac{e^{x_j/\varepsilon}}{\sum_i e^{x_i/\varepsilon}} = \mathrm{softmax}(\mathbf{x}/\varepsilon)_j$. $\square$

*Remark* 3.3. When the maximum is unique with gap $\delta := M - \max_{i:x_i \neq M} x_i > 0$, the error decays exponentially: $\mathrm{LSE}_\varepsilon(\mathbf{x}) - M = O(\varepsilon e^{-\delta/\varepsilon})$.

## 4. Method: Ultradiscretized Spiking Neurons

The key innovation of this work is applying *ultradiscretization*, a limiting procedure from tropical geometry, to derive differentiable spiking neurons. This section shows that ultradiscretization can be applied to different neural dynamics, yielding distinct models for temporal and spatial computations.

### 4.1. Ultradiscretization Framework

Ultradiscretization transforms continuous dynamical systems into max-plus (tropical) systems while preserving structural properties (Tokihiro et al., 1996). The procedure operates via the substitution $x = e^{X/\varepsilon}$ followed by the limit $\varepsilon \to 0^+$, mapping addition to $\max$ and multiplication to addition:

$$x + y = e^{X/\varepsilon} + e^{Y/\varepsilon} \to e^{\max(X,Y)/\varepsilon} \tag{4}$$

$$x \cdot y = e^{X/\varepsilon} \cdot e^{Y/\varepsilon} = e^{(X+Y)/\varepsilon} \tag{5}$$

For finite $\varepsilon > 0$, the log-sum-exp function $\mathrm{LSE}_\varepsilon$ (Eq. 3) provides a differentiable soft relaxation of the max operation. This is the foundation for all ultradiscretized neurons presented here.

#### 4.1.1. TEMPORAL DYNAMICS: ULTRALIF

UltraLIF is derived from the standard single-neuron LIF ordinary differential equation. The LIF model is a linear RC-circuit simplification of the Hodgkin-Huxley model (Hodgkin & Huxley, 1952), which describes full conductance-based ionic channel dynamics; LIF retains the passive membrane decay while abstracting away active conductances. The membrane potential evolves according to:

$$\tau_m \frac{dv}{dt} = -(v - v_{\mathrm{rest}}) + R \cdot I(t) \tag{6}$$

where $\tau_m$ is the membrane time constant, $v_{\mathrm{rest}}$ the resting potential, $R$ the membrane resistance, and $I(t)$ the input current.

Applying forward Euler discretization with timestep $\Delta t$ and setting $v_{\mathrm{rest}} = 0$ yields:

$$v^{(t+1)} = \underbrace{\left( 1 - \frac{\Delta t}{\tau_m} \right)}_{=:\tau_0} v^{(t)} + I^{(t)} \tag{7}$$

where the leak factor $\tau_0 \in (0, 1)$ in the standard decaying regime (the derivation below sets $T := \log \tau_0 < 0$); Definition 4.1 relaxes this to $\tau_0 > 0$ for the learnable variant.

The ultradiscretization transform (Tokihiro et al., 1996) proceeds by substituting $v = e^{V/\varepsilon}$, $I = e^{J/\varepsilon}$, and parameterizing the leak as $\tau = e^{T/\varepsilon}$ where $T = \log \tau_0 < 0$:

$$e^{V^{(t+1)}/\varepsilon} = e^{T/\varepsilon} \cdot e^{V^{(t)}/\varepsilon} + e^{J^{(t)}/\varepsilon} = e^{(V^{(t)}+T)/\varepsilon} + e^{J^{(t)}/\varepsilon} \tag{8}$$

Taking $\varepsilon \cdot \log$ of both sides and the limit $\varepsilon \to 0^+$ recovers the max-plus dynamics:

$$V^{(t+1)} = \max \left( V^{(t)} + T, J^{(t)} \right)$$
$$= \max \left( V^{(t)} + \log \tau_0, J^{(t)} \right) \tag{9}$$

For differentiable training, the hard maximum is relaxed to the log-sum-exp for finite $\varepsilon > 0$:

$$V_\varepsilon^{(t+1)} = \mathrm{LSE}_\varepsilon \left( V^{(t)} + \log \tau_0, I^{(t)} \right) \tag{10}$$

This **2-term LSE** captures temporal membrane integration with learnable leak. As $\varepsilon \to 0^+$, $\mathrm{LSE}_\varepsilon \to \max$ and $\sigma((V-\theta)/\varepsilon) \to H(V-\theta)$, recovering the hard tropical LIF (Eq. 9); the coupling $\tau = e^{T/\varepsilon} \to 0$ in this limit (since $T < 0$). Crucially, in the soft formulation (Eq. 10), $\log \tau_0$ appears as a plain additive shift independent of $\varepsilon$: the $\varepsilon$-coupling present in the derivation is absorbed by the $\varepsilon \cdot \log(\cdot)$ transformation, so $\tau_0$ and $\varepsilon$ can be optimized independently. **UltraPLIF** exploits this by making $\tau_0$ an additional learnable parameter (via $\log \tau_0 \in \mathbb{R}$).

### 4.1.2. SPATIAL DYNAMICS: ULTRADLIF

An analogous derivation applies ultradiscretization to spatial dynamics, capturing lateral interactions across a neuronal population. Consider a simplified diffusive coupling where membrane potentials spread locally:

$$\frac{\partial v}{\partial t} = D\nabla^2 v \tag{11}$$

where $D > 0$ is the diffusion coefficient. This models gap junction (electrical synapse) coupling, where ionic currents flow directly between neurons enabling voltage spread (Connors & Long, 2004; Spek et al., 2020). The diffusion equation provides a first-order approximation of such lateral interactions.

Discretizing the Laplacian via finite differences $\nabla^2 v \approx (v_{i-1} - 2v_i + v_{i+1})/\Delta x^2$ and applying forward Euler in time yields:

$$v_i^{(t+1)} = v_i^{(t)} + \frac{D\Delta t}{\Delta x^2}(v_{i-1}^{(t)} - 2v_i^{(t)} + v_{i+1}^{(t)}) \tag{12}$$

At the balanced diffusion regime where $D\Delta t/\Delta x^2 = 1/3$, this simplifies to uniform spatial averaging where each neuron and its neighbors contribute equally:

$$v_i^{(t+1)} = \frac{1}{3}v_{i-1}^{(t)} + \frac{1}{3}v_i^{(t)} + \frac{1}{3}v_{i+1}^{(t)} \tag{13}$$

This choice of $1/3$ lies within the von Neumann stability bound for explicit finite difference schemes applied to the 1D diffusion equation, which requires $D\Delta t/\Delta x^2 \leq 1/2$ for numerical stability. The value $1/3$ ensures stability while providing symmetric treatment of a neuron and its immediate neighbors, a natural balance for lateral coupling.

**Remark on subtraction.** Standard ultradiscretization cannot handle subtraction directly, as there is no tropical analog of $x - y$ in the max-plus semiring (Ochiai & Nacher, 2005). This limitation is circumvented by selecting $\alpha = 1/3$: expanding Eq. (12) gives coefficients $\alpha$, $(1-2\alpha)$, $\alpha$ for the three terms, and at $\alpha = 1/3$ all become $1/3 > 0$, eliminating subtraction entirely (Eq. (13)). For more general diffusion regimes, inversible max-plus algebras extend the framework to handle subtraction via $x - y \to \max(X, Y + \eta)$, where $\eta$ is an inverse element (Ochiai & Nacher, 2005).

Applying the ultradiscretization transform with $v = e^{V/\varepsilon}$ and taking $\varepsilon \to 0^+$:

$$V_i^{(t+1)} = \max\left(V_{i-1}^{(t)}, V_i^{(t)}, V_{i+1}^{(t)}\right) \tag{14}$$

This limit corresponds to *morphological dilation*, a max-pooling operation over the spatial neighborhood.

Relaxing the hard maximum to the log-sum-exp for finite $\varepsilon > 0$ gives:

$$V_{i,\varepsilon}^{(t+1)} = \mathrm{LSE}_\varepsilon\left(V_{i-1}^{(t)}, V_i^{(t)}, V_{i+1}^{(t)}\right) \tag{15}$$

This **3-term LSE** captures lateral spatial smoothing across neurons. External input $I_i^{(t)}$ is added separately (Eq. 19), following the standard neural field convention where diffusion handles lateral coupling and an additive term represents external drive.

### 4.1.3. COMPARISON OF DERIVATIONS

*Table 1.* Ultradiscretization applied to temporal and spatial dynamics.

|  | UltraLIF | UltraDLIF |
|---|---|---|
| Source | LIF ODE | Diffusion PDE |
| Equation | $dv/dt = -v/\tau_m + I$ | $\partial v/\partial t = D\nabla^2 v$ |
| LSE terms | 2 (temporal) | 3 (spatial) |
| Soft form | $\mathrm{LSE}(V + \log \tau_0, I)$ | $\mathrm{LSE}(V_{-1}, V_0, V_{+1})$ |
| Models | Membrane decay | Lateral diffusion |

Both models share the same theoretical foundation (ultradiscretization, LSE soft relaxation) but capture different biological phenomena. UltraLIF models single-neuron temporal dynamics; UltraDLIF models population-level spatial interactions.

### 4.2. Neuron Models

Building on the ultradiscretization framework, complete neuron models are defined incorporating spike generation and reset mechanisms. Both variants share the same spike and reset logic, differing only in membrane dynamics.

**Definition 4.1** (UltraLIF Neuron (Temporal)). For temperature $\varepsilon > 0$, leak factor $\tau_0 > 0$, and threshold $\theta > 0$:

$$\tilde{V}_\varepsilon^{(t+1)} = \mathrm{LSE}_\varepsilon\left(V_\varepsilon^{(t)} + \log \tau_0, I^{(t)}\right) \tag{16}$$

$$s_\varepsilon^{(t+1)} = \sigma\left(\frac{\tilde{V}_\varepsilon^{(t+1)} - \theta}{\varepsilon}\right) \tag{17}$$

$$V_\varepsilon^{(t+1)} = \tilde{V}_\varepsilon^{(t+1)} \cdot (1 - s_\varepsilon^{(t+1)}) + V_{\mathrm{reset}} \cdot s_\varepsilon^{(t+1)} \tag{18}$$

where $\sigma(z) = (1 + e^{-z})^{-1}$ is the logistic sigmoid and $V_{\mathrm{reset}} = 0$.

**Definition 4.2** (UltraDLIF Neuron (Spatial))**.** For temperature $\varepsilon > 0$, threshold $\theta > 0$, and neuron index $i$:

$$\tilde{V}_{i,\varepsilon}^{(t+1)} = \text{LSE}_\varepsilon\left(V_{i-1,\varepsilon}^{(t)}, V_{i,\varepsilon}^{(t)}, V_{i+1,\varepsilon}^{(t)}\right) + I_i^{(t)} \quad (19)$$

$$s_{i,\varepsilon}^{(t+1)} = \sigma\left(\frac{\tilde{V}_{i,\varepsilon}^{(t+1)} - \theta}{\varepsilon}\right) \quad (20)$$

$$V_{i,\varepsilon}^{(t+1)} = \tilde{V}_{i,\varepsilon}^{(t+1)} \cdot (1 - s_{i,\varepsilon}^{(t+1)}) + V_{\text{reset}} \cdot s_{i,\varepsilon}^{(t+1)} \quad (21)$$

The LSE operates over the spatial neighborhood (circular boundary conditions).

**Soft Spike Mechanism.** The soft spike $s_\varepsilon \in (0,1)$ interpolates between no-spike ($s_\varepsilon \approx 0$) and spike ($s_\varepsilon \approx 1$). This differs fundamentally from surrogate gradient methods: surrogate methods use a hard $H(V - \theta)$ in the forward pass but a smooth $g'(V)$ in the backward pass, whereas UltraLIF uses the *same* smooth $\sigma((V - \theta)/\varepsilon)$ in both passes.

**On Soft vs. Binary Spikes.** A natural concern is that $s_\varepsilon \in (0,1)$ does not represent "true" binary spikes. However, this deviation from binary behavior is not problematic in practice. The soft spike is used during training for gradient computation, while at inference one can employ small $\varepsilon$ or hard thresholding ($s = H(V - \theta)$) for neuromorphic deployment, following a standard training-inference separation analogous to dropout or batch normalization. Furthermore, the output layer uses mean spike rate $\hat{y} = \frac{1}{T}\sum_t s^{(t)}$, which is inherently robust to soft versus hard individual spikes since the classification decision depends on aggregate activity rather than precise spike values. Most importantly, Proposition 5.2 guarantees that $s_\varepsilon \to H(V - \theta)$ as $\varepsilon \to 0$, providing a principled path from soft training dynamics to binary inference behavior.

**Learnable Parameters.** The temperature $\varepsilon$ is made learnable via $\varepsilon = \exp(\log \varepsilon_{\text{param}})$, initialized to $\varepsilon_0 = 1.0$. For UltraLIF, the leak factor $\tau_0$ can also be learned (UltraPLIF variant). For UltraDLIF, similarly UltraDPLIF learns $\tau_0$ for an optional temporal component. During training, the network discovers optimal soft-to-hard trade-offs, implementing automatic curriculum learning.

### 4.3. Network Architecture

A feedforward SNN with $L$ layers of UltraLIF neurons is constructed as:

$$I_l^{(t)} = W_l \cdot s_{l-1}^{(t)} + b_l \quad (22)$$

$$V_l^{(t+1)} = \text{LSE}_\varepsilon\left(V_l^{(t)} + \log \tau_0, I_l^{(t)}\right)(1 - s_l^{(t)})$$

$$+ V_{\text{reset}} \cdot s_l^{(t)} \quad (23)$$

$$s_l^{(t+1)} = \sigma\left((V_l^{(t+1)} - \theta)/\varepsilon\right) \quad (24)$$

where $l \in \{1, \ldots, L\}$ indexes layers. The output layer employs spike rate coding:

$$\hat{y} = \frac{1}{T}\sum_{t=1}^{T} s_L^{(t)} \quad (25)$$

## 5. Theoretical Analysis

### 5.1. Convergence to Tropical LIF Dynamics

**Lemma 5.1** (Sigmoid Convergence)**.** *Let* $\sigma_\varepsilon(x) := \sigma(x/\varepsilon)$. *For* $x \neq 0$, $\lim_{\varepsilon \to 0^+} \sigma_\varepsilon(x) = H(x)$ *with exponential convergence rate* $|\sigma_\varepsilon(x) - H(x)| \leq e^{-|x|/\varepsilon}$.

*Proof.* For $x > 0$: $\sigma_\varepsilon(x) = (1 + e^{-x/\varepsilon})^{-1} \to 1$ as $\varepsilon \to 0^+$.

For $x < 0$: $\sigma_\varepsilon(x) = e^{x/\varepsilon}/(e^{x/\varepsilon} + 1) \to 0$ as $\varepsilon \to 0^+$.

The error bound follows from $|1 - \sigma_\varepsilon(x)| = e^{-x/\varepsilon}/(1 + e^{-x/\varepsilon}) \leq e^{-x/\varepsilon} = e^{-|x|/\varepsilon}$ for $x > 0$. For $x < 0$: $|\sigma_\varepsilon(x)| = 1/(1 + e^{-x/\varepsilon}) \leq e^{x/\varepsilon} = e^{-|x|/\varepsilon}$. $\square$

**Proposition 5.2** (Convergence to Tropical LIF Dynamics)**.** *Let* $\{V_\varepsilon(t)\}$ *denote the UltraLIF trajectory with temperature* $\varepsilon > 0$, *and* $\{v(t), s(t)\}$ *the tropical LIF trajectory (Eq. 9, the max-plus limit) with* $V_{\text{reset}} = 0$. *Assume (A1) bounded inputs* $|I(t)| \leq I_{\max}$ *and (A2) threshold margin* $\delta_t := |v(t) - \theta| > 0$. *Then for each fixed* $t$, $\lim_{\varepsilon \to 0^+} V_\varepsilon(t) = v(t)$ *and* $\lim_{\varepsilon \to 0^+} s_\varepsilon(t) = s(t)$, *with:*

$$|V_\varepsilon(t) - v(t)| \leq t\varepsilon \log 2 + tCe^{-\delta_{\min}/\varepsilon},$$

$$|s_\varepsilon(t) - s(t)| \leq e^{-\delta_{\min}/\varepsilon}$$

*where* $\delta_{\min} := \min_{k \leq t} \delta_k > 0$ *(finite minimum over a fixed LIF trajectory, independent of* $\varepsilon$*) and* $C$ *depends only on* $I_{\max}$ *and* $\tau_0$. *The correction* $tCe^{-\delta_{\min}/\varepsilon}$ *is exponentially dominated by* $t\varepsilon \log 2$ *as* $\varepsilon \to 0^+$*; the bound holds for each fixed* $t$ *(not uniformly in* $t$*). The set of inputs violating (A2) has Lebesgue measure zero.*

*Proof.* By strong induction on $t$. Base case: $V_\varepsilon(0) = v(0)$; spike convergence by Lemma 5.1. For the inductive step, the update $F_\varepsilon(V) = \tilde{V}(1 - s_\varepsilon) + V_{\text{reset}} \cdot s_\varepsilon$ with $\tilde{V} = \text{LSE}_\varepsilon(V + \log \tau_0, I)$ satisfies: (i) no spike ($s_\varepsilon \to 0$): $F_\varepsilon \to \max(v(t) + \log \tau_0, I(t)) = v(t+1)$; (ii) spike ($s_\varepsilon \to 1$): $F_\varepsilon \to V_{\text{reset}} = 0$, matching LIF reset. The error decomposes as $V_\varepsilon(t+1) - v(t+1) = (\tilde{V}_\varepsilon(t+1) - v(t+1)) - \tilde{V}_\varepsilon(t+1) \cdot s_\varepsilon(t+1)$. By 1-Lipschitz of $\text{LSE}_\varepsilon$ (Lemma C.1), the first term is bounded by $t\varepsilon \log 2 + \varepsilon \log 2$. The second term satisfies $|\tilde{V}_\varepsilon(t+1)| \cdot s_\varepsilon(t+1) \leq C e^{-\delta_{\min}/\varepsilon}$ under (A1)–(A2), which is absorbed into the $e^{-\delta_{\min}/\varepsilon}$ accumulation. Full details in Appendix C. $\square$

**Corollary 5.3** (UltraDLIF Convergence)**.** *The analogous result holds for UltraDLIF: as* $\varepsilon \to 0^+$*, the 3-term* $\text{LSE}_\varepsilon(V_{i-1}, V_i, V_{i+1}) \to \max(V_{i-1}, V_i, V_{i+1})$ *by*

*Lemma 3.2 with n=3, and the full UltraDLIF trajectory converges to the max-plus diffusion dynamics (Eq. 14) with error bound $|V_{i,\varepsilon}(t) - V_i(t)| \leq t \cdot \varepsilon \log 3$.*

## 5.2. Gradient Properties

**Proposition 5.4** (Bounded Spike Gradients). *For any $\varepsilon > 0$, consider the single-step spike output $s_\varepsilon = \sigma((\tilde{V}_\varepsilon - \theta)/\varepsilon)$ where $\tilde{V}_\varepsilon$ is the pre-reset voltage. The voltage-level gradient satisfies $0 < \frac{\partial s_\varepsilon}{\partial \tilde{V}_\varepsilon} \leq \frac{1}{4\varepsilon}$ for all finite $\tilde{V}_\varepsilon$. For weights $W$ with normalized input $x \in [0,1]^d$ at a single layer: $\left|\frac{\partial s_\varepsilon}{\partial W_{ij}}\right| \leq \frac{1}{4\varepsilon}$. These bounds apply to the gradient of $s_\varepsilon$ with respect to pre-reset quantities; the full BPTT gradient additionally includes a reset-path term $-\tilde{V}_\varepsilon \cdot \frac{\partial s_\varepsilon}{\partial \tilde{V}_\varepsilon}$ that is bounded whenever $\tilde{V}_\varepsilon$ is bounded.*

*Proof.* Let $z = (\tilde{V}_\varepsilon - \theta)/\varepsilon$. Then $\frac{\partial s_\varepsilon}{\partial \tilde{V}_\varepsilon} = \frac{\sigma(z)(1-\sigma(z))}{\varepsilon}$. The function $\sigma(1 - \sigma)$ achieves maximum $1/4$ at $\sigma = 1/2$, establishing the upper bound. Positivity follows since $\sigma(z) \in (0,1)$ for finite $z$. For the weight gradient, the chain rule gives $\frac{\partial s_\varepsilon}{\partial W_{ij}} = \frac{\partial s_\varepsilon}{\partial \tilde{V}_\varepsilon} \cdot \frac{\partial \tilde{V}_\varepsilon}{\partial I_j} \cdot x_j$, where $\frac{\partial \tilde{V}_\varepsilon}{\partial I_j} \leq 1$ (softmax component of the LSE gradient) and $|x_j| \leq 1$ for normalized inputs, giving $|\frac{\partial s_\varepsilon}{\partial W_{ij}}| \leq \frac{1}{4\varepsilon}$. Note that $\frac{\partial s_\varepsilon}{\partial W_{ij}} = 0$ when $x_j = 0$; non-vanishing holds at the voltage level but not necessarily for individual weights under sparse inputs. $\square$

**Corollary 5.5** (Gradient Scaling). *The temperature $\varepsilon$ controls the bias-variance trade-off in gradients: larger $\varepsilon$ yields smaller gradients and a smoother optimization landscape, while smaller $\varepsilon$ yields larger gradients near threshold and a better LIF approximation.*

**Corollary 5.6** (Dead Neuron Resistance). *For a fixed surrogate sharpness $\varepsilon_0$ and pre-reset membrane potential magnitude $M := \mathbb{E}[|\tilde{V}_\varepsilon - \theta|]$ with $M \gg \varepsilon_0$, the surrogate gradient decays exponentially: $\frac{\partial s}{\partial \tilde{V}} \leq \frac{1}{\varepsilon_0}e^{-M/\varepsilon_0} \to 0$ (using $\sigma(z)(1-\sigma(z)) \leq e^{-z}$ for $z > 0$). UltraLIF's learnable $\varepsilon$ avoids this: empirically, $\varepsilon$ scales with input complexity, ranging from 0.58 on MNIST (784-dim) to 2.24 on CIFAR-10 (3072-dim, Table 17), tracking effective membrane magnitude at each layer. For deeper architectures with Batch Normalization (BN), such as ResNet50 with 2048-dim features, fixed $\varepsilon$ saturates: gradients leave the $1/(4\varepsilon)$ peak (Proposition 5.4) for the flat sigmoid tails and vanish, while learnable $\varepsilon$ adapts to keep neurons near the peak, as confirmed by the ResNet50 result (Table 7: LIF 31.83% vs. Ultra 92.78%). This failure mode is distinct from the skip-connection incompatibility in fully-spiking networks addressed by SEW-ResNet (Fang et al., 2021a); this finding concerns BN-scale mismatch at the spiking head of hybrid ANN–SNN architectures, which SEW-ResNet's architectural modifications do not address.*

**Proposition 5.7** (Scaling Robustness of Learnable $\varepsilon$). *Let $M := \mathbb{E}[|\tilde{V} - \theta|]$ be the mean membrane deviation from*

*threshold. If $\varepsilon$ is fixed at $\varepsilon_0 \ll M$, the surrogate gradient satisfies $|\partial s/\partial \tilde{V}|_{\varepsilon_0} \leq \frac{1}{\varepsilon_0}e^{-M/\varepsilon_0}$. If $\varepsilon$ is learnable and converges to $\varepsilon^* = \Theta(M)$, then at typical $\tilde{V} = \theta + M$, letting $\kappa := \sigma(1)(1-\sigma(1)) \approx 0.197$:*

$$\left|\frac{\partial s}{\partial \tilde{V}}\right|_{\varepsilon^*} = \frac{\kappa}{\varepsilon^*} = \Theta\left(\frac{1}{M}\right). \tag{26}$$

*The ratio of learnable to fixed gradient grows as $\Theta(e^{M/\varepsilon_0}/M)$, exponentially in $M$.*

*Proof.* The fixed-$\varepsilon_0$ bound is Corollary 5.6. For the learnable case, write $\varepsilon^* = cM$ for constant $c > 0$. At $\tilde{V} = \theta + M$, $z^* = (\tilde{V} - \theta)/\varepsilon^* = 1/c$, giving $|\partial s/\partial \tilde{V}|_{\varepsilon^*} = \sigma(1/c)(1 - \sigma(1/c))/\varepsilon^* = \Theta(1/M)$. Dividing by the fixed-$\varepsilon_0$ bound yields the ratio $\Theta(\varepsilon_0 e^{M/\varepsilon_0}/M) = \Theta(e^{M/\varepsilon_0}/M)$. That learned $\varepsilon$ converges to $\Theta(M)$ is supported empirically: $\varepsilon$ scales from 0.58 on MNIST (784-dim) to 2.24 on CIFAR-10 (3072-dim, Table 17). $\square$

## 5.3. Forward-Backward Consistency

*Remark* 5.8 (Gradient Consistency). For any $\varepsilon > 0$, Ultra-LIF is a composition of smooth operations ($\text{LSE}_\varepsilon$, sigmoid, affine maps), so the chain rule applies exactly: the backward pass differentiates the same function computed forward. Surrogate methods use $H(V-\theta)$ forward but differentiate a surrogate $g(V-\theta)$ backward; Gygax & Zenke (2025) show this can be interpreted as differentiating a stochastic forward pass with escape noise. UltraLIF avoids this mismatch: $\nabla_W \mathcal{L}(f_\varepsilon(\mathbf{x}; W))$ is an exact gradient of the actual forward computation.

## 5.4. Connection to Tropical Geometry

**Theorem 5.9** (Tropical Limit). *The map $D_\varepsilon : (\mathbb{R}_{\geq 0}, +, \cdot) \to (\mathbb{R}_{\max}, \oplus_\varepsilon, +)$ defined by $D_\varepsilon(x) = \varepsilon \log x$ (with $D_\varepsilon(0) := -\infty$, the additive identity of $\mathbb{R}_{\max} = \mathbb{R} \cup \{-\infty\}$), where $a \oplus_\varepsilon b = \text{LSE}_\varepsilon(a, b)$, is a semiring homomorphism. As $\varepsilon \to 0^+$, $\oplus_\varepsilon \to \oplus = \max$ (tropical addition), and UltraLIF dynamics converge to a piecewise-linear map on $\mathbb{R}_{\max}$ with polyhedral decision boundaries.*

*Proof.* For $x, y > 0$: $D_\varepsilon(x \cdot y) = \varepsilon \log(xy) = D_\varepsilon(x) + D_\varepsilon(y)$ (multiplication $\to$ addition) and $D_\varepsilon(x + y) = \varepsilon \log(x + y) = \text{LSE}_\varepsilon(D_\varepsilon(x), D_\varepsilon(y))$ (addition $\to$ softmax). The extension to $x = 0$ uses $D_\varepsilon(0) = -\infty$: $D_\varepsilon(0 \cdot y) = -\infty = -\infty + D_\varepsilon(y)$ (since $-\infty$ absorbs under $+$ in $\mathbb{R}_{\max}$) and $D_\varepsilon(0 + y) = D_\varepsilon(y) = \text{LSE}_\varepsilon(-\infty, D_\varepsilon(y))$ (since $-\infty$ is additive identity under $\oplus_\varepsilon$). Taking $\varepsilon \to 0$ yields the tropical semiring by Lemma 3.2. The limiting dynamics are max-plus compositions followed by hard thresholding, which are piecewise-linear; each linear region is a polyhedron and decision boundaries are polyhe-

*Table 2.* Test accuracy (%) on CIFAR-10. UltraPLIF (temporal) achieves best at all timesteps.

| Model | $T{=}1$ | $T{=}10$ | $T{=}30$ |
|---|---|---|---|
| LIF | 39.83 | 44.27 | 45.69 |
| PLIF | 39.83 | 45.06 | 46.15 |
| AdaLIF | 39.83 | 44.86 | 45.83 |
| FullPLIF | 39.60 | 45.43 | 46.28 |
| DSpike | 40.26 | 44.78 | 45.34 |
| DSpike+ | 40.26 | 45.42 | 46.29 |
| *Temporal (LIF ODE)* | | | |
| UltraLIF | 40.72 | 45.15 | 45.69 |
| UltraPLIF | **43.27** | **46.19** | **46.58** |
| *Spatial (Diffusion PDE)* | | | |
| UltraDLIF | 43.11 | 45.65 | 45.00 |
| UltraDPLIF | 43.11 | 45.75 | 45.74 |

dral (unions of facets of the hyperplane arrangement from Proposition C.3). These polyhedral boundaries lie within the tropical hypersurface framework of Zhang et al. (2018) when the output layer also participates in the max-plus structure; for the standard linear readout used here, the stronger tropical rationality condition does not apply, and the correct characterization is piecewise-linear with polyhedral boundaries. □

## 6. Experiments

**Setup.** Evaluation spans six benchmarks: static images (MNIST, Fashion-MNIST, CIFAR-10), neuromorphic vision (N-MNIST, DVS-Gesture), and audio (SHD). A single hidden layer with 64 neurons is used across all experiments, with timesteps $T \in \{1, 10, 30\}$. Baselines include LIF, PLIF, AdaLIF, FullPLIF, and DSpike/DSpike+ with surrogate gradients. All four ultradiscretized variants (UltraLIF, UltraPLIF, UltraDLIF, UltraDPLIF) are evaluated. An optional sparsity penalty $\mathcal{L} = \mathcal{L}_{\text{CE}} + \lambda \cdot \bar{s}$ enables explicit accuracy-efficiency trade-offs. Energy is estimated via the relative synaptic operation (SOP) count $T \cdot \bar{s}$ (Lemaire et al., 2023), which is proportional to computational energy since all models share the same architecture (Appendix D).

**Ultra-Low Latency Performance.** The ultra-low latency regime ($T{=}1$) is the most energy-critical operating point for neuromorphic deployment (Chowdhury et al., 2021): a single forward pass minimizes both inference latency and synaptic operations. UltraLIF achieves state-of-the-art at $T{=}1$ (Table 5). Crucially, **gains are largest on neuromorphic and temporal datasets**: SHD (+11.22%), DVS-Gesture (+7.96%), N-MNIST (+3.91%), and CIFAR-10 (+3.01%). On simpler static datasets, gains are smaller but consistent: Fashion-MNIST (+0.35%), MNIST (+0.09%). Among the two derivations, UltraDLIF (spatial) wins on N-MNIST, SHD, and MNIST, while UltraPLIF (temporal)

*Table 3.* Energy efficiency on CIFAR-10. Energy = relative SOP count ($T \cdot \bar{s}$), estimated from soft spike rates during training ($\bar{s} \in (0, 1)$); hard-inference binary SOP costs and accuracy are reported separately in Table 6. Sparsity penalty $\lambda$ reduces spike rate with minimal accuracy loss. At $T{=}30$, UltraPLIF with $\lambda{=}0.1$ achieves **best accuracy** while reducing energy by 50%. Full sparsity sweep ($\lambda \in \{0, 0.01, 0.1\}$, $T \in \{1, 10, 30\}$) for all models in Appendix E.3.

| Model | $T$ | Acc (%) | Spike | Energy |
|---|---|---|---|---|
| LIF | 1 | 39.83 | 0.404 | 0.40 |
| DSpike+ | 1 | 40.26 | 0.386 | 0.39 |
| UltraPLIF | 1 | 43.27 | 0.458 | 0.46 |
| UltraPLIF ($\lambda{=}0.1$) | 1 | **43.60** | **0.240** | **0.24** |
| PLIF | 10 | 45.06 | 0.356 | 3.56 |
| UltraDPLIF | 10 | 45.75 | 0.469 | 4.69 |
| UltraDPLIF ($\lambda{=}0.1$) | 10 | **45.32** | **0.338** | **3.38** |
| PLIF | 30 | 46.15 | 0.377 | 11.30 |
| UltraPLIF | 30 | 46.58 | 0.500 | 15.01 |
| UltraPLIF ($\lambda{=}0.1$) | 30 | **46.98** | **0.248** | **7.44** |

*Table 4.* Energy efficiency on MNIST. Energy = relative SOP count ($T \cdot \bar{s}$), estimated from soft spike rates during training ($\bar{s} \in (0, 1)$); hard-inference binary SOP costs and accuracy are reported separately in Table 6. UltraDLIF with $\lambda{=}0.1$ reduces spike rate by 40% ($0.446 \rightarrow 0.268$). At $T{=}10$, UltraDPLIF with $\lambda{=}0.1$ achieves 50% energy reduction. Full sparsity sweep in Appendix E.3.

| Model | $T$ | Acc (%) | Spike | Energy |
|---|---|---|---|---|
| LIF | 1 | 95.34 | 0.388 | 0.39 |
| DSpike+ | 1 | 95.58 | 0.403 | 0.40 |
| UltraDLIF | 1 | 95.67 | 0.446 | 0.45 |
| UltraDLIF ($\lambda{=}0.1$) | 1 | **95.71** | **0.268** | **0.27** |
| DSpike+ | 10 | 97.39 | 0.415 | 4.15 |
| UltraDLIF ($\lambda{=}0.01$) | 10 | **97.56** | 0.448 | 4.48 |
| UltraDPLIF ($\lambda{=}0.1$) | 10 | 97.35 | **0.237** | **2.37** |

wins on CIFAR-10, Fashion-MNIST, and DVS-Gesture (Table 18, Appendix E.1), suggesting both variants contribute complementary strengths.

**Neuromorphic Dataset Performance.** Tables 11–13 (Appendix E.1) show that ultradiscretized models dramatically outperform baselines on neuromorphic benchmarks at $T{=}1$. On N-MNIST, UltraDLIF achieves 94.14% versus 90.23% for DSpike+ (+3.91%). On DVS-Gesture, UltraPLIF (temporal) achieves the best result: 60.23% versus 52.27% for PLIF (+7.96%). These datasets capture asynchronous events from dynamic vision sensors, where the temporal structure is fundamental. The soft max-plus dynamics appear better suited to extract information from sparse, event-driven inputs than hard-thresholding surrogates.

**Sparsity-Accuracy Trade-off.** Without sparsity penalty, ultradiscretized models tend to have higher spike rates than baselines (e.g., CIFAR-10 $T{=}1$: UltraPLIF 0.458 vs. LIF 0.404), as the soft spike $s_\varepsilon \in (0, 1)$ contributes nonzero

*Table 5.* Summary: $T$=1 accuracy (%) across datasets. Best baseline and best ultradiscretized model shown in parentheses. Full per-dataset results in Appendix E.1.

| Dataset | Baseline | Best Ultra | $\Delta$ |
|---------|----------|-----------|----------|
| MNIST | 95.58 (DSpike+) | **95.67** (DLIF) | +0.09 |
| Fashion | 82.67 (DSpike+) | **83.02** (UPLIF) | +0.35 |
| CIFAR-10 | 40.26 (DSpike+) | **43.27** (UPLIF) | +3.01 |
| N-MNIST | 90.23 (DSpike+) | **94.14** (DLIF) | +3.91 |
| DVS | 52.27 (PLIF) | **60.23** (UPLIF) | +7.96 |
| SHD | 40.02 (FullPLIF) | **51.24** (DLIF) | +11.22 |

*Table 6.* Hard inference ($s$=$\mathbf{1}[V>\theta]$) at $T$=1. Best baseline uses surrogate gradients (gap = 0). Ultra retains advantage on 4/6 datasets under binary spikes.

| Dataset | Baseline | Ultra Hard | $\Delta$ |
|---------|----------|-----------|----------|
| MNIST | 95.58 (DSpike+) | 93.72 | $-1.86$ |
| Fashion | 82.67 (DSpike+) | 80.91 | $-1.76$ |
| CIFAR-10 | 40.26 (DSpike+) | **43.35** | **+3.09** |
| N-MNIST | 90.23 (DSpike+) | **92.78** | **+2.55** |
| SHD | 40.02 (FullPLIF) | **47.88** | **+7.86** |
| DVS | 52.27 (PLIF) | **56.82** | **+4.55** |

activity even below threshold. The sparsity penalty $\lambda$ addresses this and enables flexible energy-accuracy trade-offs (Tables 3, 4). On CIFAR-10, $\lambda$=0.1 reduces spike rates by 48% (0.458 $\to$ 0.240) while actually improving accuracy (43.27% $\to$ 43.60%). On MNIST, $\lambda$=0.1 reduces spike rates by 40% at $T$=1 (0.446 $\to$ 0.268) while maintaining or slightly improving accuracy. Remarkably, at $T$=30 on CIFAR-10, UltraPLIF with $\lambda$=0.1 achieves *both* best accuracy (46.98%) and 50% energy reduction compared to the no-penalty baseline.

**Timestep Scaling.** Performance gaps narrow at higher $T$, and baselines often overtake: on CIFAR-10, the $T$=1 advantage (+3.01% for UltraPLIF) persists but shrinks at $T$=10 and $T$=30. On SHD, ultradiscretized models lead dramatically at $T$=1 (+11.22%) but baselines surpass them at $T{\geq}10$. This pattern is consistent: on N-MNIST, UltraDLIF leads by +3.91% at $T$=1 but baselines lead at $T$=10. The explanation is that surrogate gradients suffer from forward-backward mismatch, but with sufficient $T$, the averaging effect of spike rate coding masks individual spike errors. At low $T$, each spike carries more information, making the consistency of ultradiscretization more valuable.

**Learnable Temperature.** The temperature $\varepsilon$ implements automatic curriculum learning, starting soft (large $\varepsilon$) for easy optimization then sharpening (small $\varepsilon$) to approximate discrete spikes. Unlike DSpike's heuristic sharpness parameter, $\varepsilon$ has principled convergence guarantees (Proposition 5.2). Ablation studies (Appendix E.2) confirm that learned $\varepsilon$ consistently outperforms fixed values. Across six datasets, $\varepsilon$ scales with input complexity: spatial models converge to

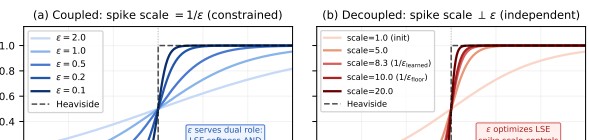

Learned spike scale enables sharper discretization while preserving LSE stability

*Figure 2.* Learned spike scale enables sharper discretization while preserving LSE stability. **(a)** In the standard formulation, spike scale = $1/\varepsilon$ is constrained: smaller $\varepsilon$ sharpens the spike but simultaneously hardens LSE dynamics. **(b)** A separate learnable spike scale $\beta \perp \varepsilon$ resolves this: $\varepsilon$ optimizes LSE membrane dynamics independently, while $\beta$ controls discretization depth. Reference lines show the standard ceiling at $1/\varepsilon_{\text{learned}}$ and $1/\varepsilon_{\text{floor}}$.

$\varepsilon \in [0.58, 2.24]$ at $T$=1 (Table 17), while temporal models converge to near-binary regimes ($\varepsilon \leq 0.23$ on 5/6 datasets), consistent with their LIF-ODE derivation. Figure 5 (Appendix E.2) visualizes this pattern.

**Spike Scale and Discretization Depth.** While $\varepsilon$ governs LSE softness, it simultaneously determines spike sharpness via $\text{sigmoid}(V/\varepsilon)$, coupling two physically distinct quantities. Under the two-noise stochastic LIF interpretation (Neftci et al., 2019), $\varepsilon$ corresponds to membrane integration noise (Gumbel scale in the LSE derivation), while spike sharpness reflects threshold noise $\sigma_\theta$, an independent source. In the standard UltraLIF formulation, the implicit spike scale = $1/\varepsilon$ is constrained by LSE stability: reducing $\varepsilon$ to sharpen spikes simultaneously hardens the LSE dynamics and impedes optimization. A decoupled variant introduces a separate learnable spike scale $\beta$ so that $s = \text{sigmoid}(\beta \cdot V)$ with $\beta \perp \varepsilon$, enabling each parameter to serve its intended role. Figure 2 illustrates this separation. Appendix G provides the formal derivation and empirical validation.

**Hard Inference.** Learned $\varepsilon \approx 0.7$–$1.1$ gives soft spikes $s_\varepsilon \in (0, 1)$ during training, so spike-rate energy estimates in Tables 3–4 reflect soft activations and overstate efficiency relative to binary neuromorphic deployment. Each trained checkpoint is evaluated with hard binary spikes ($s = \mathbf{1}[V > \theta]$) at inference time (Table 6). Surrogate-gradient baselines are unaffected (gap = 0). Hard-inference Ultra outperforms all baselines on CIFAR-10 (+3.1%), N-MNIST (+2.6%), SHD (+7.9%), and DVS-Gesture (+4.6%), with a modest gap of 2–4 pp relative to soft inference. On MNIST and Fashion-MNIST, Ultra falls slightly below the best baseline under binary spikes. The gap is narrowest for temporal models (UltraPLIF/UltraLIF), whose 2-term LSE dynamics more closely track the hard-threshold LIF trajectory.

**Computational Cost.** The LSE operation adds minor overhead ($\sim$5% wall-clock time) compared to standard LIF, but is fully parallelizable on GPU/TPU.

*Table 7.* Architecture scalability on CIFAR-10 (hidden=64, seed=42; FC rows at $T$=1, ResNet rows labeled with their timestep). Best Ultra model in parentheses. ResNet50 LIF stays dead at all $T$ (fixed threshold saturates under BN-normalized 2048-dim features); Ultra remains stable. Full $T\in\{1, 5, 10\}$ results in Appendix F.1.

| Architecture | LIF | Best Ultra ($\Delta$) |
|---|---|---|
| FC 1L | 41.79 | 46.40 DPLIF (+4.6) |
| FC 2L | 40.89 | 44.63 UPLIF (+3.7) |
| FC 3L | 40.35 | 44.15 UPLIF (+3.8) |
| Conv 2L | **74.37** | 70.54 DPLIF (−3.8) |
| RN18 $T$=1 | 93.12 | 93.37 DPLIF (+0.25) |
| RN18 $T$=10 | 93.10 | **93.50** ULIF (+0.40) |
| RN50 $T$=1 | 31.83 | **92.78** UPLIF (+61.0) |
| RN50 $T$=10 | 23.23 | **92.82** UPLIF (+69.6) |

**Architecture Scalability.** Ultra's advantage holds across architectures on CIFAR-10 (Table 7). FC depth (1L→3L): Ultra leads at every depth (+3.7–+4.6 pp), consistent with Proposition 5.4 (bounded gradients prevent the depth-induced vanishing seen in LIF). The single exception is Conv 2L, where strong spatial inductive bias (LIF 74.37%) outweighs gradient consistency. ResNet18 + spiking head: Ultra holds a small but consistent edge (+0.25 pp at $T$=1, +0.40 pp at $T$=10), with the gap growing with timesteps, consistent with per-step LSE accuracy compounding favorably at higher $T$. ResNet50 reveals a structural failure of fixed-threshold surrogates: BN-scale mismatch saturates all LIF neurons at every timestep (31.83/35.09/23.23% at $T$=1/5/10), while learnable $\varepsilon$ adapts and Ultra stays stable (∼92.8% throughout, Corollary 5.6). Full ResNet-18/50 × timestep results (also on Fashion-MNIST and N-MNIST) and FC depth tables across all 6 datasets are in Appendix F.

## 7. Discussion

**Why Spatial Dynamics Win at $T$=1.** At $T$=1, UltraLIF's memory term collapses to a 2-way competition (no prior history); UltraDLIF's 3-term LSE mixes neighboring states, amplifying cross-channel correlations that a single-step temporal model cannot access (full mechanism in Appendix B).

**Relation to Surrogate Gradients.** Unlike surrogate methods, ultradiscretized neurons use identical soft dynamics in both forward and backward passes, eliminating the gradient mismatch (Gygax & Zenke, 2025).

**Biological Interpretation.** The temperature $\varepsilon$ quantifies membrane integration noise (Gumbel scale); the decoupled spike scale $\beta$ ($= 1/\sigma_\theta$, $\sigma_\theta$ threshold noise s.d.) captures threshold noise precision (Neftci et al., 2019) — physically distinct sources (Softky & Koch, 1993) providing mechanistic grounding for treating $\varepsilon$ and $\beta$ as independent learnable parameters.

**Tropical Geometry Perspective.** Theorem 5.9 connects SNN dynamics to tropical algebraic geometry, with polyhedral decision boundaries (Zhang et al., 2018) and temporal expressivity up to $(2^h)^T$ linear regions (Appendix C.3.2).

**$\varepsilon$ Adaptation and Dead Neuron Resistance.** Table 17 reveals that learnable $\varepsilon$ tracks input complexity: spatial models converge to $\varepsilon$=0.58 on MNIST but $\varepsilon$=2.24 on CIFAR-10 and $\varepsilon$=2.17 on N-MNIST. Temporal models converge to near-binary regimes ($\varepsilon \leq 0.23$ on 5/6 datasets), consistent with the LIF-ODE derivation which directly targets hard-threshold dynamics. This adaptation prevents the dead-neuron failure (Corollary 5.6): BN-scale mismatch saturates surrogate gradients exponentially in $M/\varepsilon_0$, with the gap widening as $\Theta(e^{-M/\varepsilon_0})$ vs. $\Theta(1/M)$ for learnable $\varepsilon$ (Proposition 5.7); Table 7 confirms (LIF 31.83% vs. Ultra 92.78%).

**Limitations.** In convolutional architectures, the standard spike scale $1/\varepsilon$ appears to limit performance (−3.8 pp Conv 2L gap, Table 7); the decoupled formulation (Appendix G) addresses this theoretically, with empirical analysis in Appendix I. The soft spike $s_\varepsilon \in (0, 1)$ during training deviates from binary spikes; switching to hard inference ($s = \mathbf{1}[V > \theta]$) incurs a gap of 2–4 pp, but Ultra retains advantages over all baselines on CIFAR-10, N-MNIST, SHD, and DVS-Gesture under binary spikes (Table 6), confirming transfer to neuromorphic inference. Benchmarks are small-scale; ImageNet-scale validation remains future work.

## 8. Conclusion

UltraLIF and UltraDLIF apply ultradiscretization from tropical geometry to the LIF ODE and diffusion PDE respectively, yielding fully differentiable SNNs via log-sum-exp without surrogate gradients. Theoretical guarantees include convergence to tropical LIF dynamics, bounded gradients, and forward-backward consistency. Experiments on six benchmarks show consistent improvements over surrogate gradient baselines, with the largest gains at $T$=1 on neuromorphic and audio data (SHD +11.22%, DVS +7.96%, N-MNIST +3.91%), and learnable $\varepsilon$ avoids dead-neuron failure at scale (ResNet50: LIF 31.83% vs. Ultra 92.78%). The tropical geometry connection opens directions for principled SNN analysis, and the procedure extends naturally to richer neuron models (AdEx, Izhikevich, Hodgkin-Huxley) with the same guarantees.

## Acknowledgements

This research was supported by the Department of Education (DepEd), Philippines, under Department Order No. 013, s. 2025, which established the Education Center for AI Research (ECAIR), implemented through SEAMEO INNOTECH.

## Impact Statement

This paper presents theoretical work whose goal is to advance the field of Machine Learning, specifically in the domain of energy-efficient spiking neural networks. The ultradiscretization framework provides mathematical foundations that could accelerate deployment of neuromorphic computing systems, potentially reducing the energy footprint of machine learning applications. All experiments are validated on existing public benchmark datasets. There are many potential societal consequences of this work, none of which must be specifically highlighted here.

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

# A. Spike Mechanism Comparison

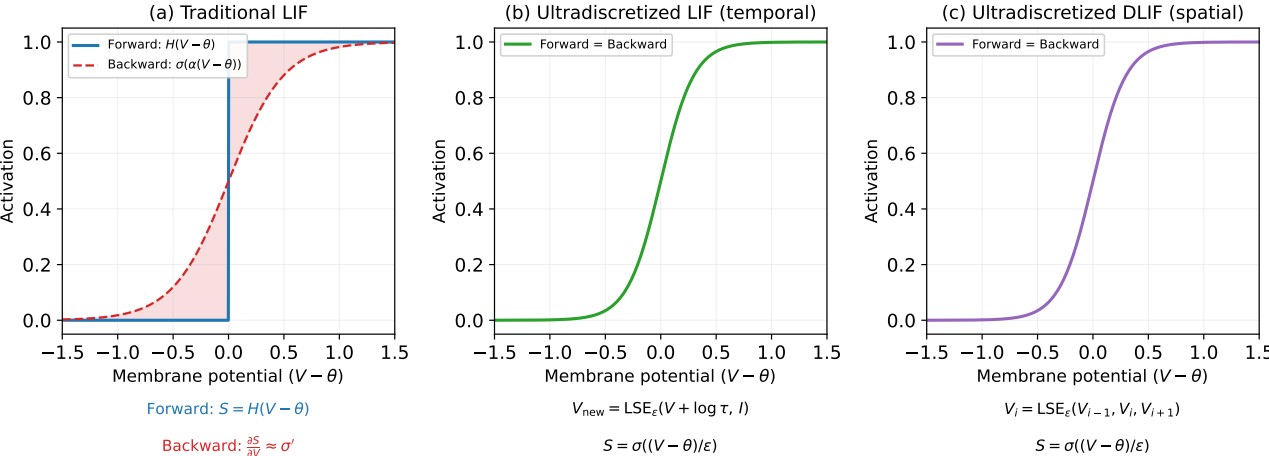

*Figure 3.* Comparison of spike mechanisms. **(a)** Traditional LIF uses Heaviside $H(V - \theta)$ in the forward pass but a smooth surrogate $\sigma'$ for gradients, creating forward-backward mismatch (shaded region). **(b)** Ultradiscretized LIF (temporal, 2-term LSE from LIF ODE) and **(c)** Ultradiscretized DLIF (spatial, 3-term LSE from diffusion PDE) use identical smooth functions in both passes, ensuring gradient consistency. The membrane potential equations below each panel show the distinct derivations.

# B. Additional Discussion

**UltraLIF Unifies Classical Activations.** The ultradiscretization framework yields a novel activation function family with deep connections to existing architectures. Observe that $\mathrm{LSE}_\varepsilon(0, x) = \varepsilon \log(1 + \exp(x/\varepsilon))$ is precisely the *softplus* function, which converges to $\mathrm{ReLU}(x) = \max(0, x)$ as $\varepsilon \to 0$. Thus, the membrane dynamics in UltraLIF generalize ReLU-family activations to the spiking domain. Meanwhile, the spike function $s_\varepsilon = \sigma((V - \theta)/\varepsilon)$ is a shifted, scaled sigmoid, i.e., a soft step function. Together, UltraLIF combines *soft-max* aggregation (ReLU family) with *soft-step* thresholding (sigmoid family), both controlled by the learnable temperature $\varepsilon$. This provides a principled spectrum of activations:

- $\varepsilon \to 0$: Hard max + hard step (classical LIF, non-differentiable)
- $\varepsilon \to \infty$: Linear + constant (no nonlinearity)
- $\varepsilon$ learned: Optimal sharpness (UltraLIF, fully differentiable)

Unlike heuristic surrogate gradients, this activation family has rigorous theoretical grounding in tropical geometry and maintains forward-backward consistency by design.

**Why Spatial Dynamics Win at $T{=}1$: Full Mechanism.** The compressed discussion in Section 7 gives the key insight; this section expands on the full mechanism. At $T{=}1$, UltraLIF's memory term $\mathrm{LSE}_\varepsilon(V^{(0)} + \log \tau_0, I)$ has no prior spike history ($V^{(0)} = \mathbf{0}$), collapsing to a 2-way competition between the decay floor ($\log \tau_0$, a fixed offset) and the current input $I$. The learnable $\tau_0$ can adapt the floor, but the structure remains a 2-term LSE regardless. UltraDLIF's 3-term LSE, by contrast, always mixes neighboring hidden neuron states regardless of prior history, pooling information across feature dimensions at every forward pass. For high-dimensional structured inputs (SHD: 700 frequency channels, N-MNIST: 2312-dim event frames), this lateral mixing amplifies cross-channel correlations that a single-step temporal model cannot access. The advantage reverses at $T \geq 10$: once UltraLIF accumulates spike history, its temporal memory becomes informative, and baselines (which also integrate over $T$ steps) close the gap (Table 13, Appendix E.1).

**Temporal vs. Spatial Variants for Spike Timing.** Tasks requiring precise spike timing may benefit differently from UltraLIF (temporal) versus UltraDLIF (spatial). UltraLIF's membrane dynamics explicitly model temporal decay and inter-timestep memory, making it better suited to tasks where exact spike order matters. UltraDLIF's lateral mixing is better suited to tasks where simultaneous cross-neuron co-activation is the relevant signal. The empirical results (SHD, DVS, N-MNIST) support this: temporal inputs at low $T$ favor UltraDLIF's lateral pooling, while at high $T$ UltraLIF's memory provides additional benefit.

**Connection to Morphological Neural Networks.** The spatial max operation in UltraDLIF (Eq. 14) corresponds to

*morphological dilation*, a fundamental operation in mathematical morphology. Deep morphological networks (Franchi et al., 2020) show that max pooling is equivalent to dilation with a flat structuring element. This connection suggests that UltraDLIF performs learned morphological operations on neural activation patterns, providing an alternative interpretation grounded in image processing theory.

# C. Proofs

## C.1. Proof of Proposition 5.2

The proof proceeds by strong induction on $t$, tracking errors explicitly at each step.

At the base case $t = 0$, the voltage error is zero by initialization ($V_\varepsilon(0) = v_0 = v(0)$), and the spike error $|s_\varepsilon(0) - s(0)| \leq e^{-\delta_0/\varepsilon} \to 0$ follows from Lemma 5.1 and the threshold margin assumption (A2).

For the inductive step, assume $|V_\varepsilon(k) - v(k)| \leq k\varepsilon \log 2 + kCe^{-\delta_{\min}/\varepsilon}$ for all $k \leq t$. The error at step $t+1$ decomposes as:

$$V_\varepsilon(t+1) - v(t+1) = \underbrace{(\tilde{V}_\varepsilon(t+1) - v(t+1))}_{\text{LSE error}} - \underbrace{\tilde{V}_\varepsilon(t+1) \cdot s_\varepsilon(t+1)}_{\text{soft-reset correction}}$$

**LSE error term.** By 1-Lipschitz of $\mathrm{LSE}_\varepsilon$ (Lemma C.1), $|\tilde{V}_\varepsilon(t+1) - v(t+1)| \leq |V_\varepsilon(t) - v(t)| + \varepsilon \log 2 \leq (t+1)\varepsilon \log 2 + tCe^{-\delta_{\min}/\varepsilon}$.

**Soft-reset correction.** When no spike occurs ($v(t+1) < \theta$), Lemma 5.1 gives $s_\varepsilon(t+1) \leq e^{-\delta_{\min}/\varepsilon}$ under (A2). Since $|\tilde{V}_\varepsilon(t+1)| \leq C'$ is bounded under (A1), this term contributes at most $C'e^{-\delta_{\min}/\varepsilon}$ per step. When a spike occurs ($v(t+1) = 0$), $s_\varepsilon(t+1) \to 1$ and $V_\varepsilon(t+1) \to 0 = v(t+1)$, so the error resets to an exponentially small value.

Combining and absorbing $C'$ into $C$: $|V_\varepsilon(t+1) - v(t+1)| \leq (t+1)\varepsilon \log 2 + (t+1)Ce^{-\delta_{\min}/\varepsilon}$. Spike convergence follows from Lemma 5.1: $|s_\varepsilon(t) - s(t)| \leq e^{-\delta_{\min}/\varepsilon}$ under (A2). $\qquad\square$

## C.2. Lipschitz Properties

**Lemma C.1.** *For fixed $\varepsilon > 0$, $\mathrm{LSE}_\varepsilon : \mathbb{R}^n \to \mathbb{R}$ is 1-Lipschitz in $\|\cdot\|_\infty$, and $\sigma_\varepsilon : \mathbb{R} \to (0,1)$ is $(4\varepsilon)^{-1}$-Lipschitz.*

*Proof.* By the mean value theorem, $|\mathrm{LSE}_\varepsilon(\mathbf{x}) - \mathrm{LSE}_\varepsilon(\mathbf{y})| \leq \sup_{\mathbf{z}} |\langle \nabla\mathrm{LSE}_\varepsilon(\mathbf{z}), \mathbf{x} - \mathbf{y}\rangle|$. Since $\nabla\mathrm{LSE}_\varepsilon = \mathrm{softmax}(\cdot/\varepsilon)$ has $\|\nabla\mathrm{LSE}_\varepsilon\|_1 = 1$ (Lemma 3.2), Hölder's inequality gives $|\langle \nabla\mathrm{LSE}_\varepsilon, \mathbf{x} - \mathbf{y}\rangle| \leq \|\nabla\mathrm{LSE}_\varepsilon\|_1 \cdot \|\mathbf{x} - \mathbf{y}\|_\infty = \|\mathbf{x} - \mathbf{y}\|_\infty$. The sigmoid bound follows from $|\sigma_\varepsilon'(x)| \leq 1/(4\varepsilon)$ (Proposition 5.4). $\qquad\square$

## C.3. Tropical Geometry Analysis

Theorem 5.9 establishes that UltraLIF dynamics converge to piecewise-linear maps on the max-plus semiring, with polyhedral decision boundaries. Three concrete extensions exploiting this structure are developed below, providing explicit expressivity bounds, temporal dynamics analysis, and capacity results.

### C.3.1. TROPICAL CHARACTERIZATION OF DECISION BOUNDARIES

An explicit geometric characterization of UltraLIF decision boundaries in the tropical limit is provided.

**Setup.** Consider a single-hidden-layer UltraLIF network with $h$ neurons, $n$ inputs, and $C$ output classes at $T=1$ in the tropical limit ($\varepsilon \to 0^+$). Starting from $V^{(0)} = \mathbf{0}$, each hidden neuron $j \in [h]$ computes:

$$V_j^{(1)} = \max(\log \tau_0, \mathbf{w}_j^\top \mathbf{x}) \tag{27}$$

where $\mathbf{w}_j \in \mathbb{R}^n$ is the weight vector for neuron $j$.

**Definition C.2** (Hyperplane Arrangement). Each neuron $j$ defines a hyperplane $\mathcal{H}_j := \{\mathbf{x} \in \mathbb{R}^n : \mathbf{w}_j^\top \mathbf{x} = \log \tau_0\}$. The collection $\mathcal{A} = \{\mathcal{H}_1, \ldots, \mathcal{H}_h\}$ partitions $\mathbb{R}^n$ into connected regions where the spike pattern $\mathbf{s} \in \{0,1\}^h$ is constant.

**Proposition C.3** (Hyperplane Arrangement Bound). *A single-hidden-layer UltraLIF partitions $\mathbb{R}^n$ into at most $R(h,n) := \sum_{k=0}^{\min(n,h)} \binom{h}{k}$ linear regions. Within each region, the network output is constant.*

*Proof.* Each neuron defines a half-space $\mathcal{H}_j^+ = \{\mathbf{x} : \mathbf{w}_j^\top \mathbf{x} > \log \tau_0\}$ where $s_j = 1$. The spike pattern is determined by membership in intersections of these half-spaces. The number of regions created by $h$ hyperplanes in $\mathbb{R}^n$ in general position (Theorem C.6) is (Zaslavsky, 1975):

$$r_n(\mathcal{A}) = \sum_{k=0}^{n} \binom{h}{k}$$

When $h < n$, terms with $k > h$ vanish since $\binom{h}{k} = 0$ for $k > h$, yielding the equivalent formula $\sum_{k=0}^{\min(n,h)} \binom{h}{k}$. Within each region, $\mathbf{s}$ is constant, so the output $\hat{y}_c = \sum_{j:s_j=1} W_{cj}^{\text{out}}$ is constant. $\square$

**Tropical Hypersurface Structure.** The decision boundary $\mathcal{B}_{ij} = \{\mathbf{x} : \hat{y}_i(\mathbf{x}) = \hat{y}_j(\mathbf{x})\}$ is a union of $(n-1)$-faces of the arrangement where class scores are equal. This realizes Theorem 5.9's connection: $\mathcal{B}_{ij}$ is a tropical hypersurface in the sense that it is the locus where two piecewise-linear functions achieve equality.

### C.3.2. TEMPORAL EXPRESSIVITY AMPLIFICATION

**Proposition C.4** (Exponential Growth). *Assume $h < n$ (hidden width less than input dimension, as in all experiments). In the tropical limit ($\varepsilon \to 0^+$), a single-hidden-layer UltraLIF unrolled for $T$ timesteps partitions $\mathbb{R}^n$ into at most $R(h,n)^T$ regions, where $R(h,n) = 2^h$.*

*Proof.* In the tropical limit, $\text{LSE}_\varepsilon \to \max$ and all spike maps are piecewise-linear. Let $\phi^{(1)} : \mathbb{R}^n \to \mathbb{R}^h$ denote the first-step map from input space to hidden membrane potentials; it is piecewise-linear with at most $R(h,n)$ regions by Proposition C.3. For $t \geq 2$, the map $\phi^{(t)} : \mathbb{R}^h \to \mathbb{R}^h$ operates on the $h$-dimensional hidden state and is piecewise-linear with at most $R(h,h)$ regions. Since $h < n$, $R(h,h) = \sum_{k=0}^{h} \binom{h}{k} = 2^h = R(h,n)$, so both per-step counts are equal.

By induction on $T$: the base case ($T = 1$) gives $R(h,n)$ regions in $\mathbb{R}^n$. For the inductive step, $\phi^{(T-1)} \circ \cdots \circ \phi^{(1)}$ has at most $R(h,n)^{T-1}$ regions in $\mathbb{R}^n$. Within each such region the map is linear, so the pullback of $\phi^{(T)}$'s $R(h,h) = R(h,n)$ hyperplanes subdivides each region into at most $R(h,n)$ subregions, giving $R(h,n)^{T-1} \cdot R(h,n) = R(h,n)^T$ total regions. $\square$

**T=1 vs T$\geq$10 Analysis.** At $T$=1, expressivity is $R(h,n) \approx 10^{19}$ for typical settings, far exceeding dataset size. The advantage comes from gradient quality: UltraLIF's forward-backward consistency avoids spurious local minima induced by surrogate gradient mismatch.

At $T \geq 10$, expressivity grows to $R(h,n)^{10}$. Output averaging $\hat{y} = \frac{1}{T} \sum_t W^{\text{out}} \mathbf{s}^{(t)}$ smooths individual spike errors. Gradient mismatch becomes less critical as errors cancel across timesteps, explaining why baselines recover (SHD: UltraDLIF +11.22% at $T$=1 but $-2.25\%$ at $T$=30 vs best baseline (DSpike+)).

### C.3.3. ZONOTOPE VOLUME AND EXPRESSIVITY

**Definition C.5** (Zonotope). The zonotope generated by $\mathbf{w}_1, \ldots, \mathbf{w}_h \in \mathbb{R}^n$ is $\mathcal{Z}(\mathbf{w}_1, \ldots, \mathbf{w}_h) := \{\sum_{i=1}^{h} \lambda_i \mathbf{w}_i : \lambda_i \in [0,1]\}$.

The volume $\text{vol}_n(\mathcal{Z})$ quantifies geometric diversity of weight directions. When $\text{vol}_n(\mathcal{Z}) = 0$, weights are linearly dependent and the arrangement degenerates.

**Theorem C.6** (Volume-Expressivity Connection). *Let $W = [\mathbf{w}_1, \ldots, \mathbf{w}_h]^\top \in \mathbb{R}^{h \times n}$ be the weight matrix. The hyperplane arrangement achieves the maximal region count $R(h,n) = \sum_k \binom{h}{k}$ if and only if the hyperplanes are in general position. General position holds if and only if:*

1. *For any $n+1$ weight vectors $\mathbf{w}_{i_1}, \ldots, \mathbf{w}_{i_{n+1}}$, no point $\mathbf{x}$ satisfies all $n+1$ hyperplane equations simultaneously.*

2. *Any subset of $n$ weight vectors $\{\mathbf{w}_{i_1}, \ldots, \mathbf{w}_{i_n}\}$ with $n \leq \min(h, \dim(\mathbb{R}^n))$ has $\det(\mathbf{w}_{i_1}, \ldots, \mathbf{w}_{i_n}) \neq 0$ when $n = \dim(\mathbb{R}^n)$.*

*Moreover, if $h \geq n$ and the weight matrix $W$ has rank $n$, then $\text{vol}_n(\mathcal{Z}(\mathbf{w}_1, \ldots, \mathbf{w}_h)) > 0$ implies the arrangement is non-degenerate with at least $2^n$ regions.*

*Proof.* The first statement follows from Zaslavsky's characterization of general position (Zaslavsky, 1975): the arrangement achieves the maximal count when no $n+1$ hyperplanes meet at a point and all intersections are transverse. Condition (1) ensures no common intersection of $n+1$ hyperplanes. Condition (2) ensures transversality: any $n$ hyperplanes intersect at a unique point (when $n$ weight vectors are linearly independent) rather than a higher-dimensional face.

For the volume statement, suppose $h \geq n$ and $\mathrm{rank}(W) = n$. Then there exist $n$ linearly independent weight vectors, say $\mathbf{w}_{i_1}, \ldots, \mathbf{w}_{i_n}$. The zonotope contains the parallelepiped spanned by these vectors:

$$\mathcal{P} = \{\sum_{j=1}^n \lambda_j \mathbf{w}_{i_j} : \lambda_j \in [0,1]\}$$

The volume satisfies $\mathrm{vol}_n(\mathcal{Z}) \geq \mathrm{vol}_n(\mathcal{P}) = |\det(\mathbf{w}_{i_1}, \ldots, \mathbf{w}_{i_n})| > 0$ by linear independence.

These $n$ linearly independent hyperplanes partition $\mathbb{R}^n$ into at least $2^n$ regions (the $2^n$ orthants when hyperplanes pass through the origin, or their translated analogues). Thus $R \geq 2^n$ when $\mathrm{vol}_n(\mathcal{Z}) > 0$.

Conversely, if $\mathrm{vol}_n(\mathcal{Z}) = 0$, the weight vectors lie in a proper subspace of dimension $< n$, yielding a degenerate arrangement with $R < 2^n$ (at most linear in $h$). $\qquad\square$

**Corollary C.7** (Capacity Lower Bound). *For a single-hidden-layer UltraLIF with $h \geq n$ neurons and $\mathrm{rank}(W) = n$:*

$$Expressivity \geq 2^n \quad if \ \mathrm{vol}_n(\mathcal{Z}(\mathbf{w}_1, \ldots, \mathbf{w}_h)) > 0$$

**Implications.** *Initialization.* Standard random initialization (Kaiming, Xavier) samples from isotropic Gaussians, yielding near-orthogonal weights with high probability in high dimensions. This ensures $\mathrm{vol}_n(\mathcal{Z}) > 0$ and non-degenerate arrangements. Explicit orthogonalization (QR decomposition) maximizes volume for fixed norms.

*Regularization.* A volume-regularized loss $\mathcal{L} = \mathcal{L}_{\mathrm{CE}} - \alpha \log \mathrm{vol}_n(\mathcal{Z})$ encourages diverse weight directions, preventing rank collapse.

*Pruning.* If $\mathbf{w}_j \approx c \cdot \mathbf{w}_k$, removing neuron $j$ reduces $\mathrm{vol}_n(\mathcal{Z})$ negligibly, providing a principled pruning criterion based on geometric redundancy.

## D. Experimental Setup

### D.1. Datasets and Preprocessing

Evaluation is conducted on six benchmarks spanning static images, neuromorphic vision, and audio:

**Static Image Datasets.**

- **MNIST**: $28 \times 28$ grayscale handwritten digits, 10 classes. 60,000 train / 10,000 test samples. Normalization: mean 0.1307, std 0.3081.

- **Fashion-MNIST**: $28 \times 28$ grayscale fashion items, 10 classes. 60,000 train / 10,000 test samples. Normalization: mean 0.2860, std 0.3530.

- **CIFAR-10**: $32 \times 32$ RGB natural images, 10 classes. 50,000 train / 10,000 test samples. Normalization per channel: mean (0.4914, 0.4822, 0.4465), std (0.247, 0.243, 0.262). Training augmentation: random crop ($32 \times 32$ with 4-pixel padding), random horizontal flip.

**Neuromorphic Datasets.**

- **N-MNIST**: Neuromorphic MNIST captured with DVS camera. 60,000 train / 10,000 test samples. Input dimension: $2 \times 34 \times 34$ (ON/OFF polarity channels).

- **DVS-Gesture**: 11 hand gesture classes recorded with DVS128 camera. 1,176 train / 288 test samples. Input dimension: $2 \times 128 \times 128$.

**Audio Dataset.**

- **SHD**: Spiking Heidelberg Digits, 20 spoken digit classes. 8,156 train / 2,264 test samples. Input dimension: 700 frequency channels.

**Temporal Encoding.** For static datasets, rate coding converts pixel intensities to spike trains:

$$P(\text{spike at } t) = \text{gain} \cdot x_{\text{pixel}}, \quad \text{gain} = 0.5 \tag{28}$$

where $x_{\text{pixel}} \in [0, 1]$ is the normalized pixel value. For neuromorphic datasets, events are binned into $T$ temporal frames using the Tonic library's `ToFrame` transform with `n_time_bins` $= T$.

## D.2. Model Hyperparameters

*Table 8.* Hyperparameters for all neuron models.

| Parameter | Value | Description |
|---|---|---|
| *Common parameters* | | |
| $\theta$ (threshold) | 0.5 | Spike threshold |
| $\tau_0$ (leak) | 0.9 | Membrane time constant |
| $V_{\text{reset}}$ | 0.0 | Reset potential |
| *Surrogate gradient (baselines)* | | |
| $\beta$ (sharpness) | 10.0 | Sigmoid surrogate steepness |
| *AdaLIF specific* | | |
| $\beta_{\text{adapt}}$ | 0.1 | Threshold adaptation strength |
| $\tau_{\text{adapt}}$ | 0.9 | Adaptation decay constant |
| *DSpike specific* | | |
| $b_0$ (init) | 4.0 | Initial sharpness parameter |
| *Ultradiscretized models* | | |
| $\varepsilon_0$ (init) | 1.0 | Initial temperature |
| $\varepsilon$ range | [0.1, 20.0] | Clamped during training |

## D.3. Architecture

Single hidden layer with 64 neurons. Input-to-hidden and hidden-to-output are fully connected layers. Timesteps $T \in \{1, 10, 30\}$ to evaluate performance across temporal regimes. Output is computed as mean spike rate over time:

$$\hat{y} = \frac{1}{T} \sum_{t=1}^{T} W_{\text{out}} \cdot s^{(t)} \tag{29}$$

## D.4. Baseline Model Definitions

All baselines use surrogate gradients: hard spike $s = H(v - \theta)$ in the forward pass, smooth gradient $\partial s / \partial v = \sigma'((v - \theta)\beta)$ in the backward pass.

**LIF** (Leaky Integrate-and-Fire):

$$v^{(t+1)} = \tau v^{(t)} + I^{(t)} \tag{30}$$

$$s^{(t+1)} = H(v^{(t+1)} - \theta), \quad v \leftarrow v(1 - s) \tag{31}$$

**PLIF** (Parametric LIF, Fang et al., 2021b):

$$v^{(t+1)} = \tau v^{(t)} + I^{(t)}, \quad \tau = \sigma(\tau_{\text{param}}) \text{ learnable} \tag{32}$$

**AdaLIF** (Adaptive LIF, Bellec et al., 2018):

$$v^{(t+1)} = \tau v^{(t)} + I^{(t)} \tag{33}$$

$$\theta^{(t+1)} = \theta_0 + \beta_{\text{adapt}} \cdot b^{(t)} \tag{34}$$

$$b^{(t+1)} = \tau_{\text{adapt}} \cdot b^{(t)} + (1 - \tau_{\text{adapt}}) \cdot s^{(t)} \tag{35}$$

where $b$ is the adaptation variable that increases after spikes.

**FullPLIF** (Fully Parametric LIF):

$$v^{(t+1)} = \tau v^{(t)} + I^{(t)} \tag{36}$$

$$\tau = \sigma(\tau_{\text{param}}), \quad \theta = \sigma(\theta_{\text{param}}) \tag{37}$$

Both $\tau$ and $\theta$ are learnable, constrained to $(0, 1)$ via sigmoid.

**DSpike** (Li et al., 2021):

$$v^{(t+1)} = \tau v^{(t)} + I^{(t)} \tag{38}$$

$$s^{(t+1)} = \frac{\tanh(k(v_{\text{norm}} - 0.5)) + \tanh(k/2)}{2\tanh(k/2)} \tag{39}$$

where $v_{\text{norm}} = v/(2\theta)$ normalizes membrane potential to $[0, 1]$, and $k$ is a learnable sharpness parameter (initialized to 4.0).

**DSpike+**: DSpike with learnable $\tau = \sigma(\tau_{\text{param}})$.

### D.5. Proposed Methods

Ultradiscretized variants use soft spike in both forward and backward passes (no surrogate gradients):

**UltraLIF** (Temporal, 2-term LSE from LIF ODE):

$$V^{(t+1)} = \text{LSE}_\varepsilon(V^{(t)} + \log\tau_0, \ I^{(t)}) \tag{40}$$

$$s_\varepsilon^{(t+1)} = \sigma((V^{(t+1)} - \theta)/\varepsilon) \tag{41}$$

**UltraDLIF** (Spatial, 3-term LSE from diffusion PDE):

$$V_i^{(t+1)} = \text{LSE}_\varepsilon(V_{i-1}^{(t)}, V_i^{(t)}, V_{i+1}^{(t)}) + I_i^{(t)} \tag{42}$$

$$s_{i,\varepsilon}^{(t+1)} = \sigma((V_i^{(t+1)} - \theta)/\varepsilon) \tag{43}$$

UltraPLIF and UltraDPLIF add learnable $\tau = \sigma(\tau_{\text{param}})$.

### D.6. Sparsity Penalty

To encourage energy efficiency, ultradiscretized variants support an optional sparsity penalty:

$$\mathcal{L} = \mathcal{L}_{\text{CE}} + \lambda \cdot \bar{s} \tag{44}$$

where $\bar{s}$ is the mean spike rate and $\lambda \in \{0, 0.01, 0.1\}$. This enables explicit control over the accuracy-efficiency trade-off.

### D.7. Training Details

**Loss Function.** Cross-entropy on mean spike rates:

$$\mathcal{L}_{\text{CE}} = -\sum_c y_c \log(\text{softmax}(\hat{y})_c) \tag{45}$$

*Table 9.* Training configuration.

| Parameter | Value |
|---|---|
| Optimizer | Adam |
| Learning rate | $10^{-3}$ |
| Batch size | 128 |
| Epochs | 100 |
| LR scheduler | Cosine annealing |
| Weight init | PyTorch default (Kaiming) |
| Seed | 42 |

**Energy Estimation.** SNN energy consumption is estimated using the standard synaptic operation (SOP) framework (Lemaire et al., 2023). In SNNs, spike-driven computation replaces multiply-accumulate (MAC) operations with accumulate-only (AC) operations, since pre-synaptic activations are binary:

$$E_{\text{SNN}} = T \cdot \bar{s} \cdot N_{\text{syn}} \cdot E_{\text{AC}} \tag{46}$$

$$E_{\text{ANN}} = N_{\text{syn}} \cdot E_{\text{MAC}} \tag{47}$$

where $T$ is the number of timesteps, $\bar{s}$ is the mean spike rate, $N_{\text{syn}}$ is the number of synaptic connections, $E_{\text{AC}} \approx 0.9\,\text{pJ}$, and $E_{\text{MAC}} \approx 4.6\,\text{pJ}$ at 45nm technology (Horowitz, 2014). Since all models in this work share the same architecture (and thus the same $N_{\text{syn}}$), the energy column in results tables reports the **relative SOP count** $T \cdot \bar{s}$, which is proportional to $E_{\text{SNN}}$ up to a constant factor. This enables direct energy comparison across models. Note that this estimate focuses on computational energy and does not account for memory access or data movement overhead, which can be significant in practice (Yan et al., 2024).

### D.8. Hardware and Compute

All experiments were conducted on NVIDIA T4 GPUs (16GB VRAM) using PyTorch 2.0+ with `torch.compile` for acceleration. Training time per model:

- MNIST/Fashion: ∼3 minutes (100 epochs)

- CIFAR-10: ∼8 minutes (100 epochs)

- N-MNIST: ∼15 minutes (100 epochs)

- DVS-Gesture: ∼20 minutes (100 epochs)

- SHD: ∼10 minutes (100 epochs)

Total compute for all experiments: approximately 50 GPU-hours across three T4 VMs.

**Note on neuromorphic hardware.** All experiments in this work were conducted on conventional GPUs. Deployment on dedicated neuromorphic hardware (e.g., Intel Loihi 2, IBM TrueNorth, SpiNNaker 2, BrainScaleS-2) has not yet been evaluated. Since the ultradiscretized spike function converges to hard thresholding as $\varepsilon \to 0$ (Proposition 5.2), the trained models can in principle be mapped to neuromorphic substrates by quantizing the soft spike to binary at inference time. Benchmarking latency, energy consumption, and accuracy on neuromorphic platforms is an important direction for future work.

## E. Additional Experiments

### E.1. Per-Dataset Full Results

Full accuracy tables for MNIST, N-MNIST, DVS-Gesture, and SHD across all timesteps. CIFAR-10 is in Table 2 (main body); Fashion-MNIST is in Table 18.

*Table 10.* Test accuracy (%) on MNIST. UltraDLIF (spatial) achieves best at $T{=}1$; UltraPLIF (temporal) at $T{=}30$.

| Model | $T{=}1$ | $T{=}10$ | $T{=}30$ |
|---|---|---|---|
| LIF | 95.34 | **97.45** | 97.45 |
| PLIF | 95.34 | 97.40 | 97.33 |
| AdaLIF | 95.34 | 97.43 | 97.45 |
| FullPLIF | 95.27 | 97.33 | 97.31 |
| DSpike | 95.58 | 97.38 | 97.48 |
| DSpike+ | 95.58 | 97.39 | 97.27 |
| *Temporal (LIF ODE)* | | | |
| UltraLIF | 94.37 | 97.14 | 97.46 |
| UltraPLIF | 95.60 | 97.30 | **97.55** |
| *Spatial (Diffusion PDE)* | | | |
| UltraDLIF | **95.67** | 97.35 | 97.38 |
| UltraDPLIF | **95.67** | 97.35 | 97.40 |

*Table 11.* Test accuracy (%) on N-MNIST (neuromorphic). UltraDLIF achieves **+3.91%** over baselines at $T{=}1$.

| Model | $T{=}1$ | $T{=}10$ | $T{=}30$ |
|---|---|---|---|
| LIF | 88.54 | 97.48 | 97.29 |
| PLIF | 88.54 | 97.53 | 97.61 |
| AdaLIF | 88.54 | 97.38 | 97.30 |
| FullPLIF | 89.00 | 97.50 | 97.48 |
| DSpike | 90.23 | 97.39 | 97.59 |
| DSpike+ | 90.23 | **97.55** | 97.65 |
| *Temporal (LIF ODE)* | | | |
| UltraLIF | 90.41 | 96.10 | 95.87 |
| UltraPLIF | 93.11 | 96.33 | 95.77 |
| *Spatial (Diffusion PDE)* | | | |
| UltraDLIF | **94.14** | 97.38 | 97.46 |
| UltraDPLIF | **94.14** | 97.38 | **97.68** |

## E.2. Ablation Study: Learnable Temperature $\varepsilon$

A key design choice in ultradiscretized neurons is whether to fix the temperature parameter $\varepsilon$ or learn it during training. This ablation study on MNIST at $T{=}1$ compares fixed values $\varepsilon \in \{0.5, 1.0, 2.0\}$ against learned $\varepsilon$ (initialized to 1.0) across all four ultradiscretized variants.

**Key findings:** (1) Learned $\varepsilon$ provides consistent accuracy gains across all models, though margins are small on MNIST. (2) Smaller fixed $\varepsilon$ (0.5) outperforms larger values (2.0), consistent with Lemma 3.2: tighter approximation to max yields better LIF emulation. (3) In this 2-layer ablation architecture, learned $\varepsilon$ converges to 0.66–1.08 (Table 16); in the 1-layer main experiments, $\varepsilon$ spans a wider range and scales with input complexity across datasets (Table 17). (4) Spatial models achieve lowest spike rates with learned $\varepsilon$, suggesting the network learns to balance accuracy and efficiency.

Table 18 reports full results on Fashion-MNIST. UltraPLIF (temporal) achieves the best accuracy at $T{=}1$, while baselines lead at higher timesteps.

## E.3. Full Sparsity Results

Tables 19–24 present full sparsity results for all ultradiscretized models across sparsity penalty values $\lambda \in \{0, 0.01, 0.1\}$ and timesteps $T \in \{1, 10, 30\}$.

**Key observations across datasets:** (1) Moderate sparsity ($\lambda{=}0.1$) consistently reduces spike rates by 40–50% with minimal accuracy loss, and in several cases (MNIST $T{=}1$, Fashion $T{=}1$, CIFAR-10 $T{=}30$) actually *improves* accuracy, suggesting that sparsity acts as a regularizer. (2) The sparsity-accuracy trade-off is most favorable on temporal models (UltraPLIF), which achieve the lowest energy at competitive accuracy.

*Table 12.* Test accuracy (%) on DVS-Gesture (neuromorphic). UltraPLIF achieves **+7.96%** at $T{=}1$.

| Model | $T{=}1$ | $T{=}10$ | $T{=}30$ |
|---|---|---|---|
| LIF | 52.27 | 67.05 | **79.92** |
| PLIF | 52.27 | 68.94 | 78.79 |
| AdaLIF | 47.73 | 68.94 | 78.79 |
| FullPLIF | 47.73 | 68.56 | 77.27 |
| DSpike | 51.14 | 67.42 | 78.79 |
| DSpike+ | 51.14 | 66.67 | 78.41 |
| *Temporal (LIF ODE)* | | | |
| UltraLIF | 58.33 | **69.32** | 75.00 |
| UltraPLIF | **60.23** | 68.94 | 75.76 |
| *Spatial (Diffusion PDE)* | | | |
| UltraDLIF | 58.33 | **69.32** | 78.41 |
| UltraDPLIF | 58.33 | 68.56 | **79.92** |

*Table 13.* Test accuracy (%) on SHD (audio). At $T{=}1$, UltraDLIF achieves **+11.22%** over the best baseline (FullPLIF). Baselines lead at $T{\geq}10$.

| Model | $T{=}1$ | $T{=}10$ | $T{=}30$ |
|---|---|---|---|
| LIF | 27.69 | 72.66 | 72.66 |
| PLIF | 27.69 | 71.69 | 73.19 |
| AdaLIF | 27.69 | **74.03** | 71.07 |
| FullPLIF | 40.02 | 71.69 | 72.48 |
| DSpike | 38.21 | 72.04 | 70.89 |
| DSpike+ | 38.21 | 72.75 | **73.85** |
| *Temporal (LIF ODE)* | | | |
| UltraLIF | 44.88 | 58.79 | 59.14 |
| UltraPLIF | 46.91 | 57.73 | 59.45 |
| *Spatial (Diffusion PDE)* | | | |
| UltraDLIF | **51.24** | 67.62 | 71.60 |
| UltraDPLIF | **51.24** | 68.90 | 67.84 |

## F. Architecture Depth Comparison

All runs: hidden=64, sparsity=0, seed=42, 100 epochs. **2L** and **3L** denote 2-layer and 3-layer fully connected networks. Model order within each group: LIF, UltraDLIF, UltraDPLIF, UltraLIF, UltraPLIF.

**Key depth findings.** (1) **T=1 robustness on SHD**: LIF collapses dramatically at depth ($-18.4$ pp at 2L, $-16.2$ pp at 3L); UltraDLIF degrades only $-2.9$ pp at 2L. (2) **T=10 at 3L**: CIFAR-10 is the only dataset where Ultra retains its advantage (+1.89 pp for UltraPLIF); all others show per Proposition 5.2 that compounding $\varepsilon \log 2$ error over timesteps (bound $O(t\varepsilon \log 2)$, growing linearly with $T$) eventually erodes the advantage. (3) N-MNIST at 2L T=10: LIF leads at 97.72%; spatial Ultra is close (UltraDPLIF 97.58%, UltraDLIF 97.49%), consistent with near-ceiling saturation.

### F.1. ResNet Backbone Results

ResNet18/50 backbone with a spiking UltraLIF classification head (hybrid ANN–SNN), across timesteps $T{\in}\{1,5,10\}$. Accuracy (%); best per column in bold. On ResNet50, LIF collapses to dead neurons at *every* timestep (BN-scale saturation under 2048-dim features), while all Ultra variants remain stable; this matches the Corollary 5.6 prediction independent of $T$.

## G. Spike Scale and Discretization Depth

### G.1. Two-Noise Stochastic LIF Derivation

The standard stochastic LIF model posits two independent noise sources (Neftci et al., 2019): (1) *membrane integration noise*, fluctuations in the input current $I$ governing membrane accumulation, and (2) *threshold noise*, fluctuations in the

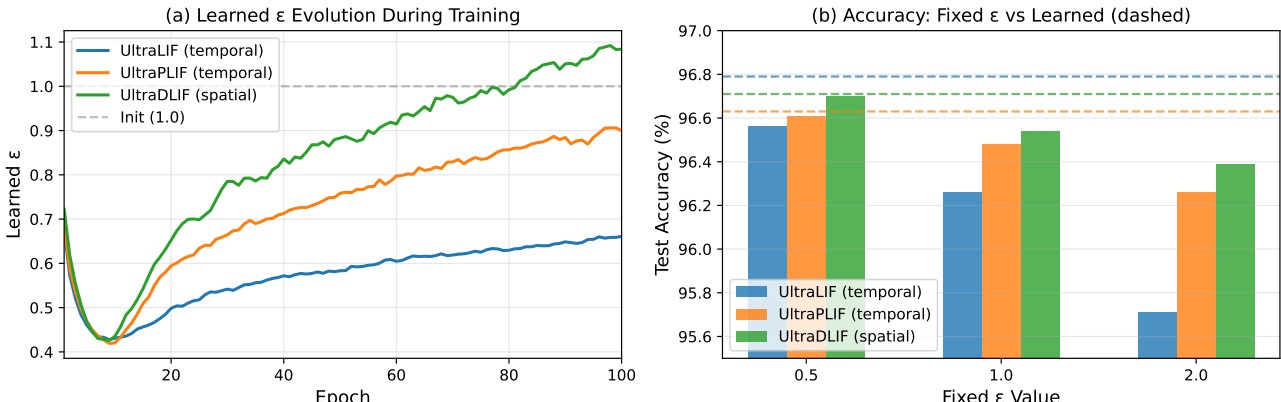

*Figure 4.* Epsilon ablation on MNIST ($T$=1, 100 epochs). (a) Learned $\varepsilon$ exhibits a characteristic U-shaped trajectory: initial drop from 1.0 to $\sim$0.42 (sharpening phase), followed by recovery to model-specific optima (0.66–1.08). This suggests the network first learns sharp discrimination, then softens for generalization. (b) Learned $\varepsilon$ (dashed lines) consistently matches or exceeds all fixed values across models, validating the benefit of learnable temperature.

*Table 14.* Ablation: Effect of learnable $\varepsilon$ on accuracy (%). Learned $\varepsilon$ consistently achieves best or tied-best accuracy across all models.

| Model | $\varepsilon$=0.5 | $\varepsilon$=1.0 | $\varepsilon$=2.0 | Learned |
|---|---|---|---|---|
| *Temporal (LIF ODE)* | | | | |
| UltraLIF | 96.56 | 96.26 | 95.71 | **96.79** |
| UltraPLIF | 96.61 | 96.48 | 96.26 | **96.63** |
| *Spatial (Diffusion PDE)* | | | | |
| UltraDLIF | 96.70 | 96.54 | 96.39 | **96.71** |
| UltraDPLIF | 96.70 | 96.54 | 96.39 | **96.71** |

firing threshold $\theta$ governing spike emission.

Under the max-plus algebra derivation of UltraLIF, these correspond to distinct parameters. The LSE membrane update

$$V_{t+1} = \mathrm{LSE}_\varepsilon(\log \tau + V_t, \ I_t) \tag{48}$$

arises from Gumbel-distributed perturbations on the input current with scale $\varepsilon$ (Tokihiro et al., 1996). Independently, the sigmoid spike function $s = \mathrm{sigmoid}(V/\sigma_\theta)$ arises from Gaussian threshold noise with standard deviation $\sigma_\theta$, where $1/\sigma_\theta$ is the threshold precision.

In the *coupled* UltraLIF formulation, both roles are collapsed onto $\varepsilon$: the spike becomes $s = \mathrm{sigmoid}(V/\varepsilon)$, so $\varepsilon$ simultaneously controls LSE softness (membrane noise scale) and spike sharpness (threshold precision $1/\varepsilon$). This coupling is not principled: smaller $\varepsilon$ sharpens spikes as desired, but simultaneously hardens LSE dynamics, reducing gradient magnitude through the membrane update and impeding optimization.

The *decoupled* formulation introduces $\beta = 1/\sigma_\theta$ as a learnable parameter independent of $\varepsilon$:

$$V_{t+1} = \mathrm{LSE}_\varepsilon(\log \tau + V_t, \ I_t) \tag{49}$$
$$s_t = \mathrm{sigmoid}(\beta \cdot V_t) \tag{50}$$

Both $\varepsilon > 0$ and $\beta > 0$ are parameterized as $\exp(\cdot)$ and optimized via gradient descent.

**Proposition G.1** (Decoupled Gradient Paths). *In the decoupled formulation* (49)–(50), $\partial\mathcal{L}/\partial\beta$ *does not depend on* $\varepsilon$: *it flows entirely through the spike path* $s_t = \mathrm{sigmoid}(\beta V_t)$ *with no $\varepsilon$ term. The gradient* $\partial\mathcal{L}/\partial\varepsilon$ *carries a $\beta$-dependent factor* $\beta \cdot s_t(1-s_t)$ *through the spike gates; however, the membrane gradient* $\partial V_t/\partial\varepsilon$—*the informative signal for $\varepsilon$—is $\beta$-independent, determined solely by* (49). *Hence $\varepsilon$ and $\beta$ receive gradient signals from structurally distinct computational paths.*

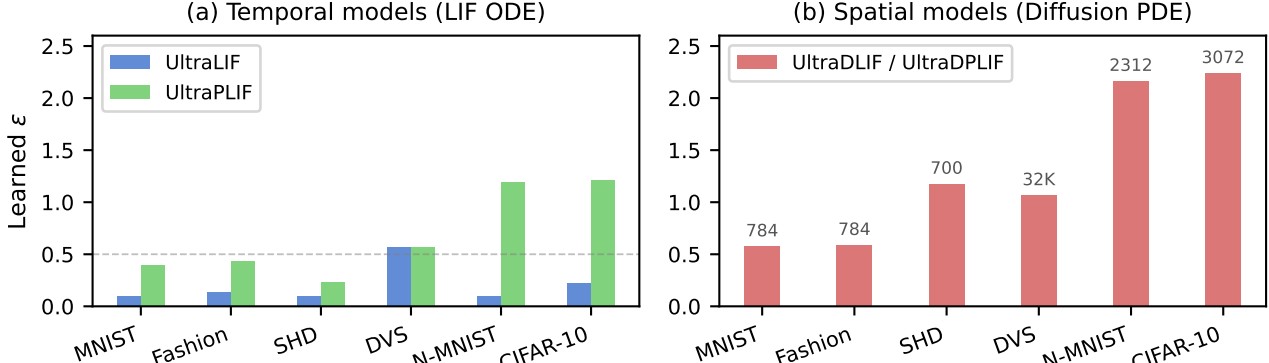

*Figure 5.* Learned $\varepsilon$ across six datasets at $T$=1 (1-layer FC, hidden=64). **(a)** Temporal models (UltraLIF, UltraPLIF) use LIF-ODE dynamics; UltraLIF converges to near-binary $\varepsilon \leq 0.57$, while UltraPLIF varies by dataset complexity. **(b)** Spatial models (UltraDLIF/UltraDPLIF) adapt $\varepsilon$ to input complexity (annotated dimension labels), ranging from 0.58 on MNIST/Fashion to 2.24 on CIFAR-10. This adaptation prevents dead-neuron failure in deeper architectures (Corollary 5.6).

*Table 15.* Ablation: Effect of learnable $\varepsilon$ on spike rate. Learned $\varepsilon$ achieves lowest spike rates for spatial models.

| Model | $\varepsilon$=0.5 | $\varepsilon$=1.0 | $\varepsilon$=2.0 | Learned |
|---|---|---|---|---|
| *Temporal (LIF ODE)* | | | | |
| UltraLIF | 0.488 | 0.579 | 0.650 | 0.500 |
| UltraPLIF | 0.427 | 0.459 | 0.528 | 0.443 |
| *Spatial (Diffusion PDE)* | | | | |
| UltraDLIF | 0.417 | 0.412 | 0.423 | **0.393** |
| UltraDPLIF | 0.417 | 0.412 | 0.423 | **0.393** |

*Proof.* $\partial\mathcal{L}/\partial\beta$ flows through the spike path $s_t = \text{sigmoid}(\beta V_t)$ only. Since $V_t$ is computed by (49) independently of $\beta$, the chain rule gives $\partial\mathcal{L}/\partial\beta = \sum_t (\partial\mathcal{L}/\partial s_t) \cdot s_t(1 - s_t) \cdot V_t$, which contains no $\varepsilon$.

$\partial\mathcal{L}/\partial\varepsilon$ flows through the membrane path (49). The chain rule gives $\partial\mathcal{L}/\partial\varepsilon = \sum_t (\partial\mathcal{L}/\partial s_t) \cdot \beta \cdot s_t(1-s_t) \cdot \partial V_t/\partial\varepsilon$. The factor $\beta \cdot s_t(1-s_t)$ scales the gradient magnitude but does not alter the direction of the membrane signal $\partial V_t/\partial\varepsilon$, which is computed from (49) independently of $\beta$. $\square$

**Proposition G.2** (Bounded Spike Scale Gradient). *For any $\varepsilon > 0$ and $\beta > 0$, $|\partial\mathcal{L}/\partial\beta| \leq \|\nabla_s\mathcal{L}\|_1 \cdot \frac{1}{4} \cdot \|V\|_\infty$, where $\frac{1}{4}$ is the maximum of $s(1-s)$ for $s \in (0,1)$.*

*Proof.* $|\partial\mathcal{L}/\partial\beta| = |\sum_t (\partial\mathcal{L}/\partial s_t) \cdot s_t(1-s_t) \cdot V_t| \leq \sum_t |(\partial\mathcal{L}/\partial s_t)| \cdot \frac{1}{4} \cdot |V_t| \leq \|\nabla_s\mathcal{L}\|_1 \cdot \frac{1}{4} \cdot \|V\|_\infty$. $\square$

The gradient remains bounded regardless of $\beta$, since larger $\beta$ sharpens $s_t$ toward $\{0,1\}$, driving $s_t(1 - s_t) \rightarrow 0$ and self-regularizing the gradient magnitude.

### G.2. Empirical Validation

Empirical results on CIFAR-10 and MNIST using Conv architectures are presented in Appendix I.

## H. SigmaLIF Ablation

To verify that the membrane update derivation (not merely a smooth spike function) drives Ultra's performance, **SigmaLIF** is introduced as a comparison baseline: standard LIF membrane dynamics ($v = \tau v + I$, surrogate LIF) with a sigmoid

*Table 16.* Final learned $\varepsilon$ values after training (initialized at 1.0).

| Model | Final $\varepsilon$ |
|---|---|
| UltraLIF (temporal) | 0.661 |
| UltraPLIF (temporal) | 0.900 |
| UltraDLIF (spatial) | 1.084 |
| UltraDPLIF (spatial) | 1.084 |

*Table 17.* Learned $\varepsilon$ values from main experiments (1-layer FC, hidden=64, seed=42, $T$=1). Spatial models (UltraDLIF/UltraDPLIF) adapt $\varepsilon$ to input complexity; temporal models (UltraLIF/UltraPLIF) converge to near-binary regimes. **Bold**: datasets where $\varepsilon > 2$.

| Dataset | UltraLIF (temporal) | UltraPLIF (temporal) | UltraDLIF (spatial) | UltraDPLIF (spatial) |
|---|---|---|---|---|
| MNIST | 0.100 | 0.397 | 0.576 | 0.576 |
| Fashion | 0.138 | 0.429 | 0.589 | 0.589 |
| SHD | 0.100 | 0.233 | 1.177 | 1.177 |
| DVS | 0.567 | 0.567 | 1.067 | 1.067 |
| CIFAR-10 | 0.227 | 1.215 | **2.240** | **2.240** |
| N-MNIST | 0.100 | 1.190 | **2.165** | **2.165** |

spike function, keeping all other hyperparameters identical. This isolates whether the max-plus membrane update provides benefits beyond using a smooth activation.

Best Ultra outperforms SigmaLIF on all three datasets (+4.4 pp SHD, +1.1 pp DVS, +1.1 pp MNIST), confirming that the principled membrane derivation from the LIF/diffusion ODE contributes independently of the smooth spike function. SigmaLIF also lacks the convergence guarantee of Proposition 5.2 and the gradient bound of Proposition 5.4.

## I. Conv Architecture Ablation: Spike Scale

Three ablation experiments are presented on the Conv architecture to characterize the role of spike scale in convolutional SNNs. All runs use hidden=64, seed=42, 100 epochs. These experiments use temporal Ultra models (UltraTLIF/UltraTPLIF) only; spatial UltraDLIF/UltraDPLIF apply 3-term LSE over neighboring neurons, overlapping with Conv's own spatial mixing, and are not evaluated here.

### I.1. Decoupled Spike Scale Ablation

To test whether coupling spike sharpness to $\varepsilon$ limits performance, the decoupled-spike (DS) variant (Appendix G) is evaluated against coupled UltraLIF-Conv. The decoupled model uses $s_t = \text{sigmoid}(\beta V_t)$ with independently learned $\beta > 0$ (spike scale, sc) and $\varepsilon > 0$ (membrane softness). Results are shown in Table 30.

The learned spike scale converges to large values (sc $\approx$ 15–32 on MNIST and Fashion, sc $\approx$ 8–15 on CIFAR), confirming that the decoupled model can independently sharpen spikes toward the LIF Heaviside limit. Yet the performance gap persists, particularly on CIFAR-10 ($-11$ to $-18$ pp). This rules out spike-sharpness coupling as the cause and points instead to the membrane dynamics themselves.

### I.2. Batch Normalization as an Input-Scale Adapter

Having ruled out spike sharpness (DS ablation), the hypothesis that the gap arises from input-scale mismatch in the LSE membrane update is tested. BN is inserted after each convolutional layer (Conv $\to$ BN $\to$ Pool $\to$ Neuron), applied identically to LIF and all Ultra variants; all five BN models are evaluated on CIFAR-10 ($T$=1, $T$=10), MNIST ($T$=1), and Fashion-MNIST ($T$=1).

**Proposition I.1** (LSE Gradient Scale Sensitivity). *Let $h \sim \mathcal{N}(\mu, \sigma^2)$ be a scalar conv feature and let $\tilde{V} = V[t-1] + \log \tau$ be the decayed prior state (treated as fixed). For the Ultra membrane update $V[t] = \text{LSE}_\varepsilon(\tilde{V}, h)$, the chain rule gives $\partial V[t]/\partial h = \sigma\big((h - \tilde{V})/\varepsilon\big) =: g(h)$. The gradient variance is $\mathbb{E}[g(1 - g)]$, with $z := (h - \tilde{V})/\varepsilon \sim \mathcal{N}(\delta, \sigma^2/\varepsilon^2)$ and $\delta = (\mu - \tilde{V})/\varepsilon$. When $\sigma/\varepsilon \to \infty$, $\mathbb{E}[g(1 - g)] \to 0$ (gradient saturation); when $\sigma/\varepsilon \to 0$ with $|\delta| > 0$, $\mathbb{E}[g(1 - g)] \to 0$*

*Table 18.* Test accuracy (%) on Fashion-MNIST. UltraPLIF (temporal) achieves best at $T=1$. Baselines lead at $T\geq 10$.

| Model | $T=1$ | $T=10$ | $T=30$ |
|---|---|---|---|
| LIF | 82.45 | 86.06 | 86.70 |
| PLIF | 82.45 | 85.99 | 86.48 |
| AdaLIF | 82.45 | 86.12 | 86.59 |
| FullPLIF | 82.18 | **86.26** | 86.33 |
| DSpike | 82.67 | 86.24 | 86.42 |
| DSpike+ | 82.67 | 86.03 | **86.76** |
| *Temporal (LIF ODE)* | | | |
| UltraLIF | 81.79 | 85.76 | 86.65 |
| UltraPLIF | **83.02** | 86.03 | 86.59 |
| *Spatial (Diffusion PDE)* | | | |
| UltraDLIF | 82.79 | 85.88 | 86.01 |
| UltraDPLIF | 82.79 | 85.69 | 85.92 |

*(collapse); when $\sigma/\varepsilon = \Theta(1)$ and $\delta \approx 0$, $\mathbb{E}[g(1-g)] \approx 0.20$, bounded away from zero.*

*Proof.* The gradient follows from the LSE chain rule: $\partial\mathrm{LSE}_\varepsilon(a,b)/\partial b = e^{b/\varepsilon}/(e^{a/\varepsilon} + e^{b/\varepsilon}) = \sigma((b-a)/\varepsilon)$. Its variance is $\mathbb{E}[\sigma'(z)]$ where $z \sim \mathcal{N}(\delta, \sigma^2/\varepsilon^2)$. When $\sigma/\varepsilon \to \infty$, $\mathrm{Var}[z] \to \infty$ and $z$ concentrates near $\pm\infty$, so $g \to \mathrm{Bernoulli}(1/2)$ and $g(1-g) \to 0$ a.s.; formally $\mathbb{E}[\sigma'(z)] \to 0$ since $\sigma' \in L^1(\mathbb{R})$ and the Gaussian density spreads to zero pointwise (Riemann–Lebesgue lemma). When $\sigma/\varepsilon \to 0$, $z \to \delta$ in probability; if $|\delta| > 0$, $\mathbb{E}[g(1-g)] \to \sigma(\delta)(1-\sigma(\delta)) < 1/4$ and tends to zero as $|\delta| \to \infty$. At $\sigma/\varepsilon = 1$, $\delta = 0$: $z \sim \mathcal{N}(0,1)$ and direct quadrature gives $\mathbb{E}[\sigma'(z)] \approx 0.20$. $\qquad\square$

**Corollary I.2** (BN Stabilises Ultra Gradient Flow). *Without BN, per-channel standard deviations $\sigma_i$ vary across conv channels (empirically $\sigma_i \in [0.1, 5]$ for intermediate layers). Since $\varepsilon$ is a single learned scalar, many channels fall in the saturation regime of Proposition I.1 and contribute near-zero gradient. BN maps each channel to $\sigma_i \approx 1$ before the spiking neuron, enabling a single $\varepsilon$ to achieve well-calibrated gradient flow uniformly. Equivalently, BN places neurons in the high-sensitivity region where $g(1-g)$ approaches its maximum $1/4$ (mirroring the spike-gradient ceiling $1/(4\varepsilon)$ of Proposition 5.4) rather than the saturated tails where $g(1-g) \to 0$. LIF is naturally more robust: its surrogate gradient $\sigma'_{\mathrm{surr}}(\beta(V[t] - V_{\mathrm{th}}))$ acts on the accumulated voltage $V[t] = \sum_{s \leq t} \tau^{t-s} h[s]$, whose variance $\sigma^2/(1 - \tau^2)$ is pooled over time and whose distribution concentrates by the CLT, providing implicit scale robustness that the per-step LSE lacks.*

Results are shown in Table 31. BN reduces the LIF–Ultra gap from $\approx 11$ pp to $\approx 2$ pp on CIFAR-10 ($T=1$), and reverses it on MNIST ($-0.47$ pp $\to +0.19$ pp) and Fashion-MNIST ($-0.98$ pp $\to +0.35$ pp), confirming that the conv gap is an input-scale artefact rather than a fundamental incompatibility of LSE dynamics with spatial processing.

*Table 19.* Sparsity results on MNIST. Accuracy (%), spike rate, and relative SOP count ($T \cdot \bar{s}$).

| Model | $\lambda$ | Acc | Spike | Energy |
|---|---|---|---|---|
| *T*=1 | | | | |
| UltraLIF | 0 | 94.37 | 0.620 | 0.62 |
| UltraLIF | 0.01 | 94.55 | 0.608 | 0.61 |
| UltraLIF | 0.1 | 94.37 | 0.537 | 0.54 |
| UltraPLIF | 0 | 95.60 | 0.468 | 0.47 |
| UltraPLIF | 0.01 | 95.50 | 0.453 | 0.45 |
| UltraPLIF | 0.1 | **95.81** | 0.289 | 0.29 |
| UltraDLIF | 0 | 95.67 | 0.446 | 0.45 |
| UltraDLIF | 0.01 | 95.62 | 0.425 | 0.43 |
| UltraDLIF | 0.1 | 95.71 | 0.268 | 0.27 |
| UltraDPLIF | 0 | 95.67 | 0.446 | 0.45 |
| UltraDPLIF | 0.01 | 95.62 | 0.425 | 0.43 |
| UltraDPLIF | 0.1 | 95.71 | **0.268** | **0.27** |
| *T*=10 | | | | |
| UltraLIF | 0 | 97.14 | 0.519 | 5.19 |
| UltraLIF | 0.01 | 97.15 | 0.498 | 4.98 |
| UltraLIF | 0.1 | 97.23 | 0.328 | 3.28 |
| UltraPLIF | 0 | 97.30 | 0.492 | 4.92 |
| UltraPLIF | 0.01 | 97.28 | 0.466 | 4.66 |
| UltraPLIF | 0.1 | 97.37 | 0.242 | 2.42 |
| UltraDLIF | 0 | 97.35 | 0.479 | 4.79 |
| UltraDLIF | 0.01 | **97.56** | 0.448 | 4.48 |
| UltraDLIF | 0.1 | 97.19 | 0.254 | 2.54 |
| UltraDPLIF | 0 | 97.35 | 0.476 | 4.76 |
| UltraDPLIF | 0.01 | 97.37 | 0.444 | 4.44 |
| UltraDPLIF | 0.1 | 97.35 | **0.237** | **2.37** |
| *T*=30 | | | | |
| UltraLIF | 0 | 97.46 | 0.502 | 15.07 |
| UltraLIF | 0.01 | 97.41 | 0.475 | 14.26 |
| UltraLIF | 0.1 | 97.51 | 0.270 | 8.09 |
| UltraPLIF | 0 | **97.55** | 0.492 | 14.76 |
| UltraPLIF | 0.01 | 97.52 | 0.464 | 13.91 |
| UltraPLIF | 0.1 | 97.53 | 0.229 | 6.87 |
| UltraDLIF | 0 | 97.38 | 0.469 | 14.07 |
| UltraDLIF | 0.01 | 97.52 | 0.427 | 12.81 |
| UltraDLIF | 0.1 | 97.11 | **0.208** | **6.24** |
| UltraDPLIF | 0 | 97.40 | 0.481 | 14.43 |
| UltraDPLIF | 0.01 | 97.34 | 0.439 | 13.18 |
| UltraDPLIF | 0.1 | 96.99 | 0.209 | 6.27 |

*Table 20.* Sparsity results on Fashion-MNIST. Accuracy (%), spike rate, and relative SOP count ($T \cdot \bar{s}$).

| Model | $\lambda$ | Acc | Spike | Energy |
|---|---|---|---|---|
| $T$=1 | | | | |
| UltraLIF | 0 | 81.79 | 0.652 | 0.65 |
| UltraLIF | 0.01 | 82.06 | 0.629 | 0.63 |
| UltraLIF | 0.1 | 81.62 | 0.530 | 0.53 |
| UltraPLIF | 0 | 83.02 | 0.472 | 0.47 |
| UltraPLIF | 0.01 | 82.81 | 0.451 | 0.45 |
| UltraPLIF | 0.1 | **83.26** | 0.279 | 0.28 |
| UltraDLIF | 0 | 82.79 | 0.429 | 0.43 |
| UltraDLIF | 0.01 | 83.01 | 0.411 | 0.41 |
| UltraDLIF | 0.1 | 83.05 | 0.267 | 0.27 |
| UltraDPLIF | 0 | 82.79 | 0.429 | 0.43 |
| UltraDPLIF | 0.01 | 83.01 | 0.411 | 0.41 |
| UltraDPLIF | 0.1 | 83.05 | **0.267** | **0.27** |
| $T$=10 | | | | |
| UltraLIF | 0 | 85.76 | 0.522 | 5.22 |
| UltraLIF | 0.01 | 86.07 | 0.510 | 5.10 |
| UltraLIF | 0.1 | 85.98 | 0.334 | 3.34 |
| UltraPLIF | 0 | 86.03 | 0.493 | 4.93 |
| UltraPLIF | 0.01 | 85.93 | 0.457 | 4.57 |
| UltraPLIF | 0.1 | **86.11** | **0.273** | **2.73** |
| UltraDLIF | 0 | 85.88 | 0.456 | 4.56 |
| UltraDLIF | 0.01 | 85.63 | 0.442 | 4.42 |
| UltraDLIF | 0.1 | 85.74 | 0.292 | 2.92 |
| UltraDPLIF | 0 | 85.69 | 0.456 | 4.56 |
| UltraDPLIF | 0.01 | 85.86 | 0.427 | 4.27 |
| UltraDPLIF | 0.1 | 85.74 | 0.278 | 2.78 |
| $T$=30 | | | | |
| UltraLIF | 0 | 86.65 | 0.507 | 15.22 |
| UltraLIF | 0.01 | 86.46 | 0.485 | 14.56 |
| UltraLIF | 0.1 | 86.36 | 0.287 | 8.60 |
| UltraPLIF | 0 | 86.59 | 0.480 | 14.41 |
| UltraPLIF | 0.01 | 86.49 | 0.453 | 13.59 |
| UltraPLIF | 0.1 | **86.72** | **0.271** | **8.13** |
| UltraDLIF | 0 | 86.01 | 0.467 | 14.01 |
| UltraDLIF | 0.01 | 85.81 | 0.429 | 12.88 |
| UltraDLIF | 0.1 | 85.94 | 0.274 | 8.23 |
| UltraDPLIF | 0 | 85.92 | 0.463 | 13.88 |
| UltraDPLIF | 0.01 | 85.80 | 0.458 | 13.75 |
| UltraDPLIF | 0.1 | 85.84 | 0.276 | 8.27 |

*Table 21.* Sparsity results on CIFAR-10. Accuracy (%), spike rate, and relative SOP count ($T \cdot \bar{s}$).

| Model | $\lambda$ | Acc | Spike | Energy |
|---|---|---|---|---|
| *T=1* | | | | |
| UltraLIF | 0 | 40.72 | 0.706 | 0.71 |
| UltraLIF | 0.01 | 40.58 | 0.662 | 0.66 |
| UltraLIF | 0.1 | 39.81 | 0.459 | 0.46 |
| UltraPLIF | 0 | 43.27 | 0.458 | 0.46 |
| UltraPLIF | 0.01 | 43.22 | 0.444 | 0.44 |
| UltraPLIF | 0.1 | **43.60** | **0.240** | **0.24** |
| UltraDLIF | 0 | 43.11 | 0.481 | 0.48 |
| UltraDLIF | 0.01 | 43.13 | 0.465 | 0.47 |
| UltraDLIF | 0.1 | 43.04 | 0.337 | 0.34 |
| UltraDPLIF | 0 | 43.11 | 0.481 | 0.48 |
| UltraDPLIF | 0.01 | 43.13 | 0.465 | 0.47 |
| UltraDPLIF | 0.1 | 43.04 | 0.337 | 0.34 |
| *T=10* | | | | |
| UltraLIF | 0 | 45.15 | 0.504 | 5.04 |
| UltraLIF | 0.01 | 44.72 | 0.494 | 4.94 |
| UltraLIF | 0.1 | 45.18 | 0.311 | 3.11 |
| UltraPLIF | 0 | 46.19 | 0.494 | 4.94 |
| UltraPLIF | 0.01 | 46.06 | 0.457 | 4.57 |
| UltraPLIF | 0.1 | 46.13 | **0.241** | **2.41** |
| UltraDLIF | 0 | 45.65 | 0.471 | 4.71 |
| UltraDLIF | 0.01 | 45.58 | 0.452 | 4.52 |
| UltraDLIF | 0.1 | 45.39 | 0.334 | 3.34 |
| UltraDPLIF | 0 | 45.75 | 0.469 | 4.69 |
| UltraDPLIF | 0.01 | **46.26** | 0.452 | 4.52 |
| UltraDPLIF | 0.1 | 45.32 | 0.338 | 3.38 |
| *T=30* | | | | |
| UltraLIF | 0 | 45.69 | 0.480 | 14.39 |
| UltraLIF | 0.01 | 45.84 | 0.466 | 13.99 |
| UltraLIF | 0.1 | 45.92 | 0.285 | 8.54 |
| UltraPLIF | 0 | 46.58 | 0.500 | 15.01 |
| UltraPLIF | 0.01 | 46.31 | 0.485 | 14.56 |
| UltraPLIF | 0.1 | **46.98** | **0.248** | **7.44** |
| UltraDLIF | 0 | 45.00 | 0.491 | 14.73 |
| UltraDLIF | 0.01 | 45.32 | 0.446 | 13.38 |
| UltraDLIF | 0.1 | 45.41 | 0.322 | 9.65 |
| UltraDPLIF | 0 | 45.74 | 0.496 | 14.89 |
| UltraDPLIF | 0.01 | 46.09 | 0.450 | 13.49 |
| UltraDPLIF | 0.1 | 45.79 | 0.331 | 9.94 |

*Table 22.* Sparsity results on N-MNIST. Accuracy (%), spike rate, and relative SOP count ($T \cdot \bar{s}$).

| Model | $\lambda$ | Acc | Spike | Energy |
|---|---|---|---|---|
| *T*=1 | | | | |
| UltraLIF | 0 | 90.41 | 0.579 | 0.58 |
| UltraLIF | 0.01 | 90.37 | 0.597 | 0.60 |
| UltraLIF | 0.1 | 90.74 | 0.546 | 0.55 |
| UltraPLIF | 0 | 93.11 | 0.396 | 0.40 |
| UltraPLIF | 0.01 | 92.73 | 0.409 | 0.41 |
| UltraPLIF | 0.1 | 93.28 | 0.318 | 0.32 |
| UltraDLIF | 0 | **94.14** | 0.506 | 0.51 |
| UltraDLIF | 0.01 | 93.97 | 0.492 | 0.49 |
| UltraDLIF | 0.1 | 93.52 | 0.291 | 0.29 |
| UltraDPLIF | 0 | **94.14** | 0.506 | 0.51 |
| UltraDPLIF | 0.01 | 93.97 | 0.492 | 0.49 |
| UltraDPLIF | 0.1 | 93.52 | **0.291** | **0.29** |
| *T*=10 | | | | |
| UltraLIF | 0 | 96.10 | 0.447 | 4.47 |
| UltraLIF | 0.01 | 96.22 | 0.415 | 4.15 |
| UltraLIF | 0.1 | 96.39 | 0.222 | 2.22 |
| UltraPLIF | 0 | 96.33 | 0.445 | 4.45 |
| UltraPLIF | 0.01 | 96.34 | 0.358 | 3.58 |
| UltraPLIF | 0.1 | 96.60 | **0.127** | **1.27** |
| UltraDLIF | 0 | 97.38 | 0.460 | 4.60 |
| UltraDLIF | 0.01 | 97.37 | 0.335 | 3.35 |
| UltraDLIF | 0.1 | 97.00 | 0.155 | 1.55 |
| UltraDPLIF | 0 | 97.38 | 0.463 | 4.63 |
| UltraDPLIF | 0.01 | **97.40** | 0.306 | 3.06 |
| UltraDPLIF | 0.1 | 96.93 | 0.144 | 1.44 |
| *T*=30 | | | | |
| UltraLIF | 0 | 95.87 | 0.404 | 12.13 |
| UltraLIF | 0.01 | 95.77 | 0.391 | 11.72 |
| UltraLIF | 0.1 | 96.30 | 0.267 | 8.02 |
| UltraPLIF | 0 | 95.77 | 0.463 | 13.90 |
| UltraPLIF | 0.01 | 95.91 | 0.412 | 12.36 |
| UltraPLIF | 0.1 | 96.48 | 0.240 | 7.21 |
| UltraDLIF | 0 | 97.46 | 0.429 | 12.86 |
| UltraDLIF | 0.01 | 97.34 | 0.301 | 9.04 |
| UltraDLIF | 0.1 | 97.28 | 0.140 | 4.20 |
| UltraDPLIF | 0 | **97.68** | 0.430 | 12.89 |
| UltraDPLIF | 0.01 | 97.52 | 0.275 | 8.26 |
| UltraDPLIF | 0.1 | 97.33 | **0.125** | **3.74** |

*Table 23.* Sparsity results on DVS-Gesture. Accuracy (%), spike rate, and relative SOP count ($T \cdot \bar{s}$).

| Model | $\lambda$ | Acc | Spike | Energy |
|---|---|---|---|---|
| *T*=1 | | | | |
| UltraLIF | 0 | 58.33 | 0.726 | 0.73 |
| UltraLIF | 0.01 | 57.58 | 0.746 | 0.75 |
| UltraLIF | 0.1 | 57.95 | 0.690 | 0.69 |
| UltraPLIF | 0 | **60.23** | 0.619 | 0.62 |
| UltraPLIF | 0.01 | 57.20 | 0.610 | 0.61 |
| UltraPLIF | 0.1 | 55.68 | **0.601** | **0.60** |
| UltraDLIF | 0 | 58.33 | 0.774 | 0.77 |
| UltraDLIF | 0.01 | 56.44 | 0.833 | 0.83 |
| UltraDLIF | 0.1 | 58.71 | 0.790 | 0.79 |
| UltraDPLIF | 0 | 58.33 | 0.774 | 0.77 |
| UltraDPLIF | 0.01 | 56.44 | 0.833 | 0.83 |
| UltraDPLIF | 0.1 | 58.71 | 0.790 | 0.79 |
| *T*=10 | | | | |
| UltraLIF | 0 | 69.32 | 0.707 | 7.07 |
| UltraLIF | 0.01 | 68.56 | 0.709 | 7.09 |
| UltraLIF | 0.1 | 67.80 | 0.702 | 7.02 |
| UltraPLIF | 0 | 68.94 | 0.559 | 5.59 |
| UltraPLIF | 0.01 | 69.70 | 0.567 | 5.67 |
| UltraPLIF | 0.1 | 70.83 | **0.543** | **5.43** |
| UltraDLIF | 0 | 69.32 | 0.611 | 6.11 |
| UltraDLIF | 0.01 | 71.97 | 0.601 | 6.01 |
| UltraDLIF | 0.1 | 70.45 | 0.543 | 5.43 |
| UltraDPLIF | 0 | 68.56 | 0.615 | 6.15 |
| UltraDPLIF | 0.01 | 71.97 | 0.614 | 6.14 |
| UltraDPLIF | 0.1 | **73.11** | 0.578 | 5.78 |
| *T*=30 | | | | |
| UltraLIF | 0 | 75.00 | 0.719 | 21.58 |
| UltraLIF | 0.01 | 75.38 | 0.719 | 21.56 |
| UltraLIF | 0.1 | 75.00 | 0.707 | 21.20 |
| UltraPLIF | 0 | 75.76 | 0.593 | 17.79 |
| UltraPLIF | 0.01 | 75.76 | 0.588 | 17.63 |
| UltraPLIF | 0.1 | 76.14 | 0.559 | 16.77 |
| UltraDLIF | 0 | 78.41 | 0.560 | 16.79 |
| UltraDLIF | 0.01 | 78.41 | 0.552 | 16.56 |
| UltraDLIF | 0.1 | **81.06** | **0.501** | **15.04** |
| UltraDPLIF | 0 | 79.92 | 0.570 | 17.09 |
| UltraDPLIF | 0.01 | 77.65 | 0.556 | 16.68 |
| UltraDPLIF | 0.1 | 79.55 | 0.521 | 15.62 |

*Table 24.* Sparsity results on SHD. Accuracy (%), spike rate, and relative SOP count ($T \cdot \bar{s}$).

| Model | $\lambda$ | Acc | Spike | Energy |
|---|---|---|---|---|
| *T=1* | | | | |
| UltraLIF | 0 | 44.88 | 0.551 | 0.55 |
| UltraLIF | 0.01 | 45.76 | 0.542 | 0.54 |
| UltraLIF | 0.1 | 46.86 | 0.507 | 0.51 |
| UltraPLIF | 0 | 46.91 | 0.390 | 0.39 |
| UltraPLIF | 0.01 | 47.61 | 0.394 | 0.39 |
| UltraPLIF | 0.1 | 48.32 | **0.344** | **0.34** |
| UltraDLIF | 0 | 51.24 | 0.686 | 0.69 |
| UltraDLIF | 0.01 | 50.62 | 0.629 | 0.63 |
| UltraDLIF | 0.1 | **51.33** | 0.565 | 0.56 |
| UltraDPLIF | 0 | 51.24 | 0.686 | 0.69 |
| UltraDPLIF | 0.01 | 50.62 | 0.629 | 0.63 |
| UltraDPLIF | 0.1 | **51.33** | 0.565 | 0.56 |
| *T=10* | | | | |
| UltraLIF | 0 | 58.79 | 0.472 | 4.72 |
| UltraLIF | 0.01 | 57.99 | 0.462 | 4.62 |
| UltraLIF | 0.1 | 60.51 | 0.397 | 3.97 |
| UltraPLIF | 0 | 57.73 | 0.420 | 4.20 |
| UltraPLIF | 0.01 | 58.48 | 0.411 | 4.11 |
| UltraPLIF | 0.1 | 58.79 | **0.348** | **3.48** |
| UltraDLIF | 0 | 67.62 | 0.416 | 4.16 |
| UltraDLIF | 0.01 | 69.52 | 0.442 | 4.42 |
| UltraDLIF | 0.1 | 69.92 | 0.367 | 3.67 |
| UltraDPLIF | 0 | 68.90 | 0.461 | 4.61 |
| UltraDPLIF | 0.01 | 65.77 | 0.496 | 4.96 |
| UltraDPLIF | 0.1 | **70.27** | 0.399 | 3.99 |
| *T=30* | | | | |
| UltraLIF | 0 | 59.14 | 0.458 | 13.73 |
| UltraLIF | 0.01 | 59.41 | 0.451 | 13.52 |
| UltraLIF | 0.1 | 61.00 | 0.404 | 12.11 |
| UltraPLIF | 0 | 59.45 | 0.449 | 13.48 |
| UltraPLIF | 0.01 | 60.16 | 0.445 | 13.35 |
| UltraPLIF | 0.1 | 60.60 | **0.385** | **11.54** |
| UltraDLIF | 0 | 71.60 | 0.459 | 13.76 |
| UltraDLIF | 0.01 | 70.23 | 0.458 | 13.73 |
| UltraDLIF | 0.1 | **73.19** | 0.386 | 11.57 |
| UltraDPLIF | 0 | 67.84 | 0.454 | 13.62 |
| UltraDPLIF | 0.01 | 67.71 | 0.449 | 13.48 |
| UltraDPLIF | 0.1 | 69.88 | 0.404 | 12.13 |

*Table 25.* FC depth comparison at $T=1$ (%). Ultra wins all 6 datasets at 1L and 3L; LIF more robust at 2L only on DVS.

| Model | MNIST | Fashion | CIFAR10 | SHD | N-MNIST | DVS |
|---|---|---|---|---|---|---|
| *2 layers* | | | | | | |
| LIF | 95.97 | 82.65 | 40.89 | 19.48 | 90.01 | 52.27 |
| UltraDLIF | 96.10 | 83.43 | 44.20 | **50.84** | **94.94** | 51.52 |
| UltraDPLIF | 96.10 | 83.43 | 44.20 | **50.84** | **94.94** | 51.52 |
| UltraLIF | 95.90 | 83.07 | 43.44 | 36.09 | 90.74 | 53.41 |
| UltraPLIF | **96.22** | **83.45** | **44.63** | 42.01 | 94.10 | **56.44** |
| *3 layers* | | | | | | |
| LIF | 95.90 | 82.90 | 40.35 | 21.73 | 87.82 | 50.38 |
| UltraDLIF | 96.22 | **83.55** | 43.64 | **45.67** | **94.87** | **51.14** |
| UltraDPLIF | 96.22 | **83.55** | 43.64 | **45.67** | **94.87** | **51.14** |
| UltraLIF | 95.63 | 82.23 | 43.01 | 24.25 | 93.45 | 39.39 |
| UltraPLIF | **96.35** | 83.42 | **44.15** | 30.70 | 93.68 | 43.94 |

*Table 26.* FC depth comparison at $T{=}10$ (%). Best Ultra wins 4/6 at 2L; LIF leads on N-MNIST (near-ceiling) and MNIST.

| Model | MNIST | Fashion | CIFAR10 | SHD | N-MNIST | DVS |
|---|---|---|---|---|---|---|
| *2 layers* | | | | | | |
| LIF | **97.75** | 86.48 | 45.46 | 71.95 | **97.72** | 68.94 |
| UltraDLIF | 97.39 | 86.39 | 46.87 | **72.48** | 97.49 | 66.29 |
| UltraDPLIF | 97.40 | 86.22 | 46.65 | 70.49 | **97.58** | 64.39 |
| UltraLIF | 97.40 | 86.15 | 46.43 | 64.89 | 96.97 | 66.67 |
| UltraPLIF | 97.62 | **86.70** | **47.36** | 63.87 | 97.18 | **69.70** |
| *3 layers* | | | | | | |
| LIF | **97.83** | **86.75** | 45.56 | **73.76** | 97.51 | 68.94 |
| UltraDLIF | 97.45 | 86.31 | 46.25 | 71.11 | 97.25 | 68.94 |
| UltraDPLIF | 97.53 | 85.89 | 46.82 | 66.43 | 97.37 | 68.94 |
| UltraLIF | 97.25 | 85.86 | 46.14 | 59.23 | 96.26 | 57.95 |
| UltraPLIF | 97.52 | 86.30 | **47.45** | 60.11 | 96.59 | 67.42 |

*Table 27.* ResNet18 and ResNet50 backbone on CIFAR-10 (%).

| Model | $T{=}1$ | $T{=}5$ | $T{=}10$ |
|---|---|---|---|
| *ResNet18* | | | |
| LIF | 93.12 | 93.01 | 93.10 |
| UltraDLIF | 93.31 | **93.39** | 93.21 |
| UltraDPLIF | **93.37** | 93.33 | 93.17 |
| UltraLIF | **93.37** | 93.29 | **93.50** |
| UltraPLIF | 93.12 | 92.94 | 93.34 |
| *ResNet50* | | | |
| LIF | 31.83 | 35.09 | 23.23 |
| UltraDLIF | 92.23 | 92.41 | 92.54 |
| UltraDPLIF | 92.22 | 92.73 | 90.42 |
| UltraLIF | 91.92 | 91.85 | 91.77 |
| UltraPLIF | **92.78** | **92.88** | **92.82** |

*Table 28.* ResNet18 backbone on Fashion-MNIST and N-MNIST (%).

| Model | $T{=}1$ | $T{=}5$ | $T{=}10$ |
|---|---|---|---|
| *Fashion-MNIST* | | | |
| LIF | 93.81 | 93.65 | 93.86 |
| UltraDLIF | 93.78 | 93.66 | **94.24** |
| UltraDPLIF | 93.78 | **93.89** | 93.78 |
| UltraLIF | 93.94 | 93.81 | 93.84 |
| UltraPLIF | **93.95** | 93.69 | 93.69 |
| *N-MNIST* | | | |
| LIF | 99.20 | 99.13 | 99.13 |
| UltraDLIF | **99.23** | **99.23** | **99.23** |
| UltraDPLIF | **99.23** | **99.23** | **99.23** |
| UltraLIF | 99.19 | 99.19 | 99.19 |
| UltraPLIF | 99.22 | 99.22 | 99.22 |

*Table 29.* SigmaLIF ablation at $T{=}1$ (%). SigmaLIF uses standard LIF membrane + sigmoid spike; Ultra uses the full ultradiscretized membrane + sigmoid spike. Best Ultra outperforms SigmaLIF on all three datasets.

| Model | SHD | DVS | MNIST |
|---|---|---|---|
| SigmaLIF | 49.38 | 59.09 | 95.49 |
| UltraLIF | 44.88 | 58.33 | 94.50 |
| UltraPLIF | 46.91 | **60.23** | 95.60 |
| UltraDLIF | **53.75** | 58.33 | **96.53** |
| UltraDPLIF | **53.75** | 58.33 | **96.56** |
| Best Ultra $\Delta$ vs SigmaLIF | **+4.37** | **+1.14** | **+1.07** |

*Table 30.* Decoupled spike scale ablation (%) at $T{=}1$. $\varepsilon$ and `sc` report learned values at CIFAR-10. Despite learning sharp spikes (`sc` $\approx 8$–15 on CIFAR), Ultra-DS still trails LIF by 11–18 pp, ruling out spike-sharpness coupling as the cause of the conv gap.

| Model | MNIST $T{=}1$ | Fashion $T{=}1$ | CIFAR $T{=}1$ | $\varepsilon$ | `sc` |
|---|---|---|---|---|---|
| LIF-Conv | 99.10 | 91.15 | 74.13 | — | — |
| UltraTLIF-Conv-DS | 97.10 | 88.87 | 55.65 | 0.35 | 14.96 |
| UltraTPLIF-Conv-DS | 98.48 | 90.17 | 62.90 | 0.38 | 7.90 |

*Table 31.* BN-Conv ablation (%) at $T{=}1$. Conv $\rightarrow$ BN $\rightarrow$ Pool $\rightarrow$ Neuron. Gap row shows best Ultra $-$ LIF; negative = LIF leads. BN reduces the CIFAR-10 gap from $\approx$11 pp to $\approx$2 pp and reverses the gap on MNIST and Fashion.

| Model | MNIST $T{=}1$ | | Fashion $T{=}1$ | | CIFAR-10 $T{=}1$ | |
|---|---|---|---|---|---|---|
| | No-BN | +BN | No-BN | +BN | No-BN | +BN |
| LIF-Conv | 99.10 | 99.02 | 91.15 | 91.31 | 74.13 | 76.17 |
| UltraTLIF-Conv-BN | 95.39 | 99.15 | 89.23 | **91.66** | 55.13 | 73.01 |
| UltraTPLIF-Conv-BN | 98.63 | 99.17 | 89.81 | 91.58 | 63.31 | 74.10 |
| UltraTLIF-Conv-BN-DS | 97.10 | 99.14 | 88.87 | 91.55 | 55.65 | 73.17 |
| UltraTPLIF-Conv-BN-DS | 98.48 | **99.21** | 90.17 | 91.54 | 62.90 | 74.79 |
| UltraDLIF-Conv | **99.06** | **99.34** | 90.33 | 90.90 | 70.54 | **74.99** |
| UltraDPLIF-Conv | 99.00 | 99.29 | **90.41** | 90.94 | 70.54 | **74.99** |
| Gap (best Ultra $-$ LIF) | $-0.04$ | **+0.32** | $-0.74$ | **+0.35** | $-3.59$ | $-1.18$ |

