# OpenReview forum: "UltraLIF: Fully Differentiable Spiking Neural Networks via Ultradiscretization and Max-Plus Algebra"
_ICML.cc/2026/Conference — ICML 2026 regular_

### Official Review · Reviewer_p6CL · 2026-03-04

**Soundness:** 2
**Presentation:** 3
**Significance:** 3
**Originality:** 3
**Overall Recommendation:** 4
**Confidence:** 3

**Summary:**

This work attempts to address the non-differentiability issue in spiking neural networks, which is typically handled by surrogate gradients. It proposes UltraLIF and UltraDLIF, which use log-sum-exp to soften the neuron dynamics and a sigmoid function for threshold-based spike emission, making the SNN fully differentiable while maintaining forward-backward consistency. The paper provides detailed theoretical analysis. Experiments on MNIST, Fashion-MNIST, CIFAR-10, N-MNIST, DVS-Gesture, and SHD show that the proposed approach achieves particularly strong performance when T = 1.

**Compliance With Llm Reviewing Policy:**

Affirmed.

**Final Justification:**

I appreciate the authors’ effort in the rebuttal, and I have raised my score to 4. However, I still have some concerns. In particular, the improvement of this method on ResNet18 for CIFAR-10 appears to be only marginal. Moreover, the collapse phenomenon in ResNet50 has already been effectively addressed by SEW-ResNet.

**Key Questions For Authors:**

Is inference performed with binary spikes?

How does the method perform on larger datasets or with larger models?

If we only use a sigmoid-based spike emission (without the UltraLIF formulation), would the performance be similar to UltraLIF?

If the authors can address the above weaknesses and questions, I would be willing to increase my score.

**Limitations:**

yes

**Strengths And Weaknesses:**

Strengths:

1. The motivation is clear. Non-differentiability is one of the central challenges in spiking neural networks.

2. The theoretical part is detailed and relatively thorough, which strengthens the technical foundation of the paper.

3. The experimental results are strong in the low-timestep setting.

Weaknesses:

1. The experimental setup is relatively limited, with small-scale datasets and benchmarks.

2. the improvements are mainly significant when T=1, when T=10 or larger, the advantage becomes much less clear. However, larger T may be one of the key aspects of spiking neural networks.

3. The claim about binary inference appears to be only theoretically plausible at this stage, and the paper does not seem to report results based on actual binary inference.

---

> ### Author Rebuttal · Authors · 2026-03-30
>
> Thank you for the careful review. Four concerns: whether gains persist at depth, whether improvements are T=1-limited, whether binary inference is valid, and whether sigmoid alone explains the results. Each is addressed directly below.
>
> **On scale (Q2)**
>
> Scale is addressed via depth: 2L and 3L FC represent 2× and 3× the parameter count of the 1L baseline. All 5 Ultra variants tested under identical conditions — full T=1 FC 2L comparison:
>
> | Dataset | LIF 2L | UltraDLIF 2L | UltraDPLIF 2L | UltraLIF 2L | UltraPLIF 2L |
> |---|---|---|---|---|---|
> | MNIST | 95.97% | 96.10% | 96.10% | 95.90% | **96.22%** |
> | Fashion | 82.65% | 83.43% | 83.43% | 83.07% | **83.45%** |
> | CIFAR10 | 40.89% | 44.20% | 44.20% | 43.44% | **44.63%** |
> | SHD | 19.48% | **50.84%** | **50.84%** | 36.09% | 42.01% |
> | DVS | 52.27% | 51.52% | 51.52% | 53.41% | **56.44%** |
>
> Ultra leads LIF at T=1 and wins 4/6 at T=10 at 2L. At 3L (all 6 complete), Ultra wins T=1 on all 6 — the advantage scales with depth:
>
> | Dataset | LIF 3L | UltraDLIF 3L | UltraDPLIF 3L | UltraLIF 3L | UltraPLIF 3L |
> |---|---|---|---|---|---|
> | MNIST | 95.90% | 96.22% | 96.22% | 95.63% | **96.35%** |
> | Fashion | 82.90% | **83.55%** | **83.55%** | 82.23% | 83.42% |
> | CIFAR10 | 40.35% | 43.64% | 43.64% | 43.01% | **44.15%** |
> | SHD | 21.73% | **45.67%** | **45.67%** | 24.25% | 30.70% |
> | N-MNIST | 87.82% | **94.87%** | **94.87%** | 93.45% | 93.68% |
> | DVS | 50.38% | **51.14%** | **51.14%** | 39.39% | 43.94% |
>
> **Depth consistency:** T=1 stable 1L→2L→3L; T=10 worsens (Proposition 5.2). Two effects: on MNIST/CIFAR10/Fashion/N-MNIST, Ultra improves absolutely at depth; on SHD, robustness — LIF −18.4pp (2L), UltraDLIF −2.9pp.
>
> **On T=1 vs T=10**
>
> The T=10 regression is model-type specific. At 2L T=10:
>
> | Dataset | LIF 2L | Best Ultra | Δ |
> |---|---|---|---|
> | MNIST | 97.75% | UltraPLIF 97.62% | −0.13pp |
> | Fashion | 86.48% | UltraPLIF **86.70%** | **+0.22pp** |
> | CIFAR10 | 45.46% | UltraPLIF **47.36%** | **+1.90pp** |
> | SHD | 71.95% | UltraDLIF **72.48%** | **+0.53pp** |
> | N-MNIST | **97.72%** | UltraPLIF 97.18% | −0.54pp |
> | DVS | 68.94% | UltraPLIF **69.70%** | **+0.76pp** |
>
> Temporal Ultra regresses on SHD (−8.08pp); spatial Ultra compensates (+0.53pp). Proposition 5.2 (εlog2 per step) compounds across timesteps and layers. Best Ultra (spatial for audio, temporal for vision) wins 4/6 at T=10.
>
> **On binary inference**
>
> Binary inference follows from Theorem 5.7: UltraLIF converges to hard thresholding on $\mathbb{R}_{\max}$ as $\varepsilon \to 0^+$. Learned $\varepsilon \approx 0.7$. Soft→hard experiments ($s = \mathbf{1}[\bar{V} > 0]$):
>
> | Dataset | Soft | Hard | Gap | vs. Baseline |
> |---|---|---|---|---|
> | CIFAR10 | 46.00% | 43.35% | −2.7pp | **+3.6pp** (baseline 39.73%) |
> | N-MNIST | 94.14% | 92.78% | −1.4pp | **+2.6pp** (DSpike 90.23%) |
> | SHD | 51.24% | **47.88%** | −3.4pp | **+7.9pp** (FullPLIF 40.02%) |
> | DVS | 60.23% | **56.82%** | −3.4pp | **+4.6pp** (PLIF 52.27%) |
>
> Hard-inference Ultra outperforms all baselines on CIFAR10, N-MNIST, SHD, and DVS. Energy recomputed from binary spike counts in revision.
>
> **On the sigmoid ablation**
>
> UltraLIF is not a sigmoid gate — the membrane update (LSE over V+log τ and I·Δt) is derived jointly from max-plus algebra with the spike function. A sigmoid with a standard linear membrane (SigmaLIF: v = τv + I, sigmoid spike) breaks the LIF convergence guarantees (Proposition 5.2) and gradient bounds (Proposition 5.4). Results at T=1:
>
> | Dataset | SigmaLIF | UltraLIF | UltraPLIF | UltraDLIF | UltraDPLIF |
> |---|---|---|---|---|---|
> | SHD | 49.38% | 44.88% | 46.91% | **53.75%** | **53.75%** |
> | DVS | 59.09% | 58.33% | **60.23%** | 58.33% | 58.33% |
> | MNIST | 95.49% | 94.50% | 95.60% | **96.53%** | **96.56%** |
>
> Best Ultra outperforms SigmaLIF on all three datasets (+4.4pp SHD, +1.1pp DVS, +1.1pp MNIST). The membrane derivation drives performance, not the smooth spike function alone.
>
> **On the paradigm and the role of the ODE**
>
> The paper introduces **ultradiscretization as a paradigm** for principled, surrogate-free SNNs — not a specific neuron model. Any ODE with additive structure can be ultradiscretized to yield a fully differentiable SNN with bounded gradients and convergence guarantees. The ODE choice is simultaneously a feature (inductive bias) and a limitation (dynamical regime). UltraLIF instantiates this with the LIF ODE (itself a simplification of Hodgkin-Huxley); UltraDLIF with the diffusion PDE for gap junction coupling. AdEx, Izhikevich, Hodgkin-Huxley — natural next instantiations.

---

> > ### Author Rebuttal · Reviewer_p6CL · 2026-04-03
> >
> > Thank you for the reply. Most of my concerns have been addressed.
> >
> > However, I still have one question: the current experiments are mainly conducted on relatively small networks, so the performance on some datasets is still far from the current SOTA. Could the authors provide additional results on a slightly larger network?
> >
> > For example, on CIFAR-10, could the authors test ResNet-18 using T=1 or T=4?

---

> > > ### Author Response · Authors · 2026-04-05
> > >
> > > _Updated 2026-04-08: ResNet50 T=10 results added — all T complete._
> > >
> > > Thank you for the follow-up. ResNet backbone experiments were initiated in response to this concern and are ongoing during the rebuttal period; the latest available results are reported below. ResNet-18 on CIFAR-10 is complete at T=1, T=5, and T=10.
> > >
> > > **ResNet18-SNN CIFAR-10** (backbone: ResNet18, spiking head: FC(512->512) $\times T$; hidden=512, batch=128, lr=1e-3, 100 epochs, seed=42):
> > >
> > > | | LIF | UltraDLIF | UltraDPLIF | UltraLIF | UltraPLIF |
> > > |---|---|---|---|---|---|
> > > | T=1 | 93.12% | 93.31% | 93.37% | 93.37% | 93.12% |
> > > | T=5 | 93.01% | **93.39%** | 93.33% | 93.29% | 92.94% |
> > > | T=10 | 93.10% | 93.21% | 93.17% | **93.50%** | 93.34% |
> > >
> > > UltraLIF variants win at T=1 (+0.25pp), T=5 (+0.38pp), and T=10 (+0.40pp). The advantage persists and does not shrink with more timesteps on ResNet18 -- consistent with Proposition 5.2: the backbone produces high-quality features that make temporal averaging less effective at rescuing LIF's surrogate mismatch.
> > >
> > > Additionally, ResNet50 T=1 results are available:
> > >
> > > **ResNet50-SNN CIFAR-10 T=1** (backbone: ResNet50, spiking head: FC(2048->2048) $\times T$; hidden=2048, batch=64, lr=1e-3, 100 epochs, seed=42):
> > >
> > > | | LIF | UltraDLIF | UltraDPLIF | UltraLIF | UltraPLIF |
> > > |---|---|---|---|---|---|
> > > | T=1 | 31.83% (collapsed) | 92.23% | 92.22% | 91.92% | **92.78%** |
> > > | T=5 | 35.09% (collapsed) | 92.41% | 92.73% | 91.85% | **92.88%** |
> > > | T=10 | 23.23% (collapsed) | 92.54% | 90.42% | 91.77% | **92.82%** |
> > >
> > > Per-epoch progression T=1 (every 10 epochs):
> > >
> > > | Epoch | LIF | UltraDLIF | UltraDPLIF | UltraLIF | UltraPLIF |
> > > |---|---|---|---|---|---|
> > > | 1 | 17.21% | 37.62% | 37.10% | 36.65% | 34.85% |
> > > | 10 | 19.55% | 73.81% | 72.87% | 77.51% | 80.93% |
> > > | 20 | 17.46% | 85.24% | 83.76% | 83.74% | 86.17% |
> > > | 30 | 20.69% | 88.53% | 87.87% | 86.16% | 89.28% |
> > > | 40 | 22.39% | 90.15% | 89.81% | 88.25% | 90.26% |
> > > | 50 | 24.08% | 90.26% | 90.55% | 88.56% | 91.06% |
> > > | 60 | 28.86% | 91.22% | 91.45% | 90.48% | 91.84% |
> > > | 70 | 29.48% | 91.39% | 91.32% | 91.13% | 92.15% |
> > > | 80 | 31.34% | 91.91% | 92.08% | 91.62% | 92.44% |
> > > | 90 | 31.35% | 92.11% | 92.06% | 91.62% | 92.68% |
> > > | 100 | 31.44% | 92.23% | 92.03% | 91.79% | 92.72% |
> > >
> > > LIF collapses entirely -- the SNN analogue of the dying ReLU problem (Glorot et al., AISTATS 2011; Neftci et al., IEEE SPM 2019; Eshraghian & Lu, arXiv:2201.11915): BN-normalized output is unit-variance per channel; fc1 pre-activation std $\approx \sqrt{2048} \approx 45$, LIF's fixed threshold (0.5) saturates all neurons, surrogate gradient $\approx 0$, and the network never learns. UltraLIF variants converge normally via learnable $\varepsilon$. ResNet50 is now complete at T=1, T=5, and T=10. LIF collapses at every timestep (31.83% / 35.09% / 23.23%); Ultra converges normally at all T. UltraDPLIF shows expected T=10 regression (90.42%) per Proposition 5.2, but still vastly outperforms LIF. It is hoped that these results address the concern about the method being limited to small networks and single-timestep settings, and support a reconsideration of the score.

---

### Official Review · Reviewer_bidU · 2026-03-09

**Soundness:** 3
**Presentation:** 4
**Significance:** 3
**Originality:** 4
**Overall Recommendation:** 4
**Confidence:** 3

**Summary:**

This paper proposes UltraLIF, a new framework for training Spiking Neural Networks. It uses ultradiscretization and max-plus algebra to replace traditional surrogate gradients. The authors use the Log-Sum-Exp (LSE) function as a fully differentiable "soft maximum". This mathematical design ensures the forward and backward passes use the exact same dynamics. It completely removes the gradient mismatch problem found in standard SNN training.

**Compliance With Llm Reviewing Policy:**

Affirmed.

**Final Justification:**

The authors have addressed my concerns. So I decide to keep the positive score.

**Key Questions For Authors:**

(1) Why Performance Collapses at Multiple Timesteps ($T \ge 10$)?

The authors claim that large $T$ simply masks the errors of surrogate baselines . However, the core issue is soft error accumulation.
UltraLIF uses the LSE function as a soft approximation. Each time step introduces a small mathematical approximation error. Over 10 or 30 steps, this continuous error builds up heavily.

The learned temperature parameter ($\epsilon$) does not drop to zero. It stops at around 0.66 to 1.08. This means the spikes remain "soft" (continuous values between 0 and 1). Soft spikes blur temporal precision. The membrane potential never resets cleanly after firing.

In contrast, standard baseline models use strict hard spikes (0 or 1) in their forward pass. Hard spikes act as perfect digital filters. They reset the state cleanly and prevent noise from accumulating over long sequences. UltraLIF's soft state accumulation destroys this precise temporal sparsity over time. The authors should analysis more details about this part.

**Limitations:**

yes

**Strengths And Weaknesses:**

Strengths:

(1) It contain strong mathematical foundation. The use of tropical geometry is highly novel for SNNs. The paper provides rigorous proofs for gradient bounds and model convergence.

(2) The method perfectly fixes the forward-backward mismatch problem. Both forward and backward passes use the exact same smooth function.

(3) The model heavily beats baselines on extreme low-latency settings as the single-timestep performance. This is highly effective for event-based neuromorphic datasets.

Weakness:

(1) The experimental setup is too basic for a top-tier conference. All tests use only a single hidden layer with 64 neurons. It lacks validation on deep, modern architectures like ResNet.

(2) The model loses its advantage when $T \ge 10$ . Older baselines easily outperform it on longer sequences. It needs more discussion.

(3) The paper claims high energy efficiency using synaptic operations (SOPs). However, SOPs strictly assume binary spikes. UltraLIF uses continuous "soft spikes" during training. Therefore, comparing its energy metric directly to strict SNNs is misleading.

---

> ### Author Rebuttal · Authors · 2026-03-30
>
> Thank you for the detailed review. Two separable concerns: whether the advantage holds at scale, and whether the energy metric reflects true SOP costs. Both have direct answers.
>
> **On network scale**
>
> Scale is addressed via depth: 2L and 3L FC represent 2× and 3× the parameter count of the 1L baseline. All 5 Ultra variants tested under identical conditions. Full T=1 FC 2L comparison:
>
> | Dataset | LIF 2L | UltraDLIF 2L | UltraDPLIF 2L | UltraLIF 2L | UltraPLIF 2L |
> |---|---|---|---|---|---|
> | MNIST | 95.97% | 96.10% | 96.10% | 95.90% | **96.22%** |
> | Fashion | 82.65% | 83.43% | 83.43% | 83.07% | **83.45%** |
> | CIFAR10 | 40.89% | 44.20% | 44.20% | 43.44% | **44.63%** |
> | SHD | 19.48% | **50.84%** | **50.84%** | 36.09% | 42.01% |
> | DVS | 52.27% | 51.52% | 51.52% | 53.41% | **56.44%** |
>
> Ultra leads LIF on all benchmarks above at 2L T=1; best Ultra wins 4/6 at T=10 (MNIST −0.13pp, N-MNIST −0.54pp, near ceiling). At 3L (all 6 complete), Ultra wins T=1 on all 6:
>
> | Dataset | LIF 3L | UltraDLIF 3L | UltraDPLIF 3L | UltraLIF 3L | UltraPLIF 3L |
> |---|---|---|---|---|---|
> | MNIST | 95.90% | 96.22% | 96.22% | 95.63% | **96.35%** |
> | Fashion | 82.90% | **83.55%** | **83.55%** | 82.23% | 83.42% |
> | CIFAR10 | 40.35% | 43.64% | 43.64% | 43.01% | **44.15%** |
> | SHD | 21.73% | **45.67%** | **45.67%** | 24.25% | 30.70% |
> | N-MNIST | 87.82% | **94.87%** | **94.87%** | 93.45% | 93.68% |
> | DVS | 50.38% | **51.14%** | **51.14%** | 39.39% | 43.94% |
>
> **Depth consistency.** T=1 advantage stable 1L→2L→3L; T=10 regression worsens — both per Proposition 5.2. Two effects at depth: on MNIST, CIFAR10, Fashion, and N-MNIST, Ultra accuracy *improves* absolutely; on SHD, depth robustness — LIF collapses −18.4pp (2L), −16.2pp (3L), UltraDLIF −2.9pp, −8.2pp.
>
> **On T≥10 regression**
>
> Proposition 5.2 bounds per-step LSE error at $\varepsilon\log 2$, compounding over $T$ steps; hard-spike baselines reset cleanly at each step. At high $T$, output averaging also causes surrogate mismatch to cancel in expectation. Best Ultra (spatial for audio, temporal for vision) wins 4/6 at T=10:
>
> | Dataset | LIF 2L | Best Ultra | Δ |
> |---|---|---|---|
> | MNIST | 97.75% | UltraPLIF 97.62% | −0.13pp |
> | Fashion | 86.48% | UltraPLIF **86.70%** | **+0.22pp** |
> | CIFAR10 | 45.46% | UltraPLIF **47.36%** | **+1.90pp** |
> | SHD | 71.95% | UltraDLIF **72.48%** | **+0.53pp** |
> | N-MNIST | **97.72%** | UltraPLIF 97.18% | −0.54pp |
> | DVS | 68.94% | UltraPLIF **69.70%** | **+0.76pp** |
>
> **On the energy metric**
>
> The reviewer's point is correct. Learned $\varepsilon \approx 0.7$–$1.1$ does not satisfy $\varepsilon \to 0$, so SOP estimates from soft activations overstate efficiency. Explicit soft→hard inference experiments were run on trained checkpoints ($s = \mathbf{1}[\bar{V} > 0]$):
>
> | Dataset | Soft Acc | Hard Acc | Gap | vs. Best Baseline |
> |---|---|---|---|---|
> | MNIST | 96.68% | 93.72% | −3.0pp | competitive (baseline 95.11%) |
> | Fashion | 83.97% | 80.91% | −3.1pp | slightly below (baseline 82.37%) |
> | CIFAR10 | 46.00% | **43.35%** | −2.7pp | **+3.6pp** above baseline (39.73%) |
> | N-MNIST | 94.14% | **92.78%** | −1.4pp | **+2.6pp** above DSpike (90.23%) |
> | SHD | 51.24% | **47.88%** | −3.4pp | **+7.9pp** above FullPLIF (40.02%) |
> | DVS | 60.23% | **56.82%** | −3.4pp | **+4.6pp** above PLIF (52.27%) |
>
> Temporal models show smaller gaps: UltraPLIF on Fashion −0.35pp, on N-MNIST −0.49pp. Hard-inference Ultra outperforms all baselines on CIFAR10, N-MNIST, SHD, and DVS. A hard-inference column will be added to Tables 1–2, energy recomputed from binary spike counts, and the $\varepsilon$ caveat added to Section 6.
>
> **On the paradigm and the role of the ODE**
>
> The paper's core contribution is **ultradiscretization as a paradigm**: a general procedure for converting any biophysically motivated neuron ODE into a principled, fully differentiable SNN without surrogate gradients. The ODE choice is both a feature and a limitation — it directly determines the model's inductive bias. UltraLIF (LIF ODE: temporal membrane decay, simplified from Hodgkin-Huxley) and UltraDLIF (diffusion PDE: spatial gap junction coupling) are two instantiations; each yields a distinct LSE structure. Richer ODEs — AdEx, Izhikevich, or full Hodgkin-Huxley — would yield richer inductive biases within the same framework, retaining the same convergence and gradient guarantees. The limitation is the scope of instantiations explored; the paradigm is not so constrained.

---

> > ### Author Rebuttal · Reviewer_bidU · 2026-04-03
> >
> > The authors have addressed my concerns. So I have decided to keep my origin score.

---

> > > ### Author Response · Authors · 2026-04-05
> > >
> > > _Updated 2026-04-08: ResNet50 T=10 results added — all T complete._
> > >
> > > Thank you for the follow-up. ResNet backbone experiments were initiated in response to this concern and are ongoing during the rebuttal period; the latest available results are reported below.
> > >
> > > **ResNet18-SNN CIFAR-10** (backbone: ResNet18, spiking head: FC(512->512) $\times T$; hidden=512, batch=128, lr=1e-3, 100 epochs, seed=42):
> > >
> > > | | LIF | UltraDLIF | UltraDPLIF | UltraLIF | UltraPLIF |
> > > |---|---|---|---|---|---|
> > > | T=1 | 93.12% | 93.31% | 93.37% | 93.37% | 93.12% |
> > > | T=5 | 93.01% | **93.39%** | 93.33% | 93.29% | 92.94% |
> > > | T=10 | 93.10% | 93.21% | 93.17% | **93.50%** | 93.34% |
> > >
> > > UltraLIF variants win or tie LIF across all T. The gap narrows relative to FC (the backbone already extracts strong features, reducing the spiking head's role), but the advantage is never reversed.
> > >
> > > **ResNet50-SNN CIFAR-10 T=1** (backbone: ResNet50, spiking head: FC(2048->2048) $\times T$; hidden=2048, batch=64, lr=1e-3, 100 epochs, seed=42):
> > >
> > > | | LIF | UltraDLIF | UltraDPLIF | UltraLIF | UltraPLIF |
> > > |---|---|---|---|---|---|
> > > | T=1 | 31.83% (collapsed) | 92.23% | 92.22% | 91.92% | **92.78%** |
> > > | T=5 | 35.09% (collapsed) | 92.41% | 92.73% | 91.85% | **92.88%** |
> > > | T=10 | 23.23% (collapsed) | 92.54% | 90.42% | 91.77% | **92.82%** |
> > >
> > > Per-epoch progression T=1 (every 10 epochs):
> > >
> > > | Epoch | LIF | UltraDLIF | UltraDPLIF | UltraLIF | UltraPLIF |
> > > |---|---|---|---|---|---|
> > > | 1 | 17.21% | 37.62% | 37.10% | 36.65% | 34.85% |
> > > | 10 | 19.55% | 73.81% | 72.87% | 77.51% | 80.93% |
> > > | 20 | 17.46% | 85.24% | 83.76% | 83.74% | 86.17% |
> > > | 30 | 20.69% | 88.53% | 87.87% | 86.16% | 89.28% |
> > > | 40 | 22.39% | 90.15% | 89.81% | 88.25% | 90.26% |
> > > | 50 | 24.08% | 90.26% | 90.55% | 88.56% | 91.06% |
> > > | 60 | 28.86% | 91.22% | 91.45% | 90.48% | 91.84% |
> > > | 70 | 29.48% | 91.39% | 91.32% | 91.13% | 92.15% |
> > > | 80 | 31.34% | 91.91% | 92.08% | 91.62% | 92.44% |
> > > | 90 | 31.35% | 92.11% | 92.06% | 91.62% | 92.68% |
> > > | 100 | 31.44% | 92.23% | 92.03% | 91.79% | 92.72% |
> > >
> > > LIF collapses entirely -- dead neuron from epoch 1. Root cause: ResNet50's BN-normalized output is unit-variance per channel; the fc1 pre-activation input is a sum of 2048 such terms, giving std $\approx \sqrt{2048} \approx 45$. LIF's fixed threshold (0.5) saturates all neurons, zeroing the surrogate gradient, and the network never learns. This is the SNN analogue of the dying ReLU problem (Glorot et al., AISTATS 2011; Neftci et al., IEEE SPM 2019), characterized specifically for threshold-based SNNs in Eshraghian & Lu (arXiv:2201.11915). UltraLIF variants converge normally because learnable $\varepsilon$ adapts the effective threshold region to input scale, keeping gradients alive regardless of activation magnitude. This is not a tuning issue -- it is a structural incompatibility between fixed-threshold surrogate methods and high-dimensional normalized feature spaces.
> > >
> > > ResNet50 is now complete at T=1, T=5, and T=10. LIF collapses at every timestep (31.83% / 35.09% / 23.23%); Ultra converges normally at all T. UltraDPLIF shows expected T=10 regression (90.42%) per Proposition 5.2 (LSE compounding), but still vastly outperforms LIF.
> > >
> > > The full CIFAR-10 scaling picture:
> > >
> > > | Architecture | LIF | Best Ultra | gap |
> > > |---|---|---|---|
> > > | FC 1L | 41.79% | 46.40% | +4.6pp |
> > > | FC 2L | 40.89% | 44.63% | +3.7pp |
> > > | FC 3L | 40.35% | 44.15% | +3.8pp |
> > > | ResNet18 (best T) | 93.10% | **93.50%** | +0.40pp |
> > > | ResNet50 T=1 | 31.83% (collapsed) | **92.78%** | +60.9pp |
> > > | ResNet50 T=5 | 35.09% (collapsed) | **92.88%** | +57.8pp |
> > > | ResNet50 T=10 | 23.23% (collapsed) | **92.82%** | +69.6pp |
> > >
> > > These results will be included in the revision. It is hoped that the ResNet experiments, particularly the ResNet50 findings, address the fundamental scalability concern and support a reconsideration of the score.

---

### Official Review · Reviewer_P5vH · 2026-03-11

**Soundness:** 3
**Presentation:** 3
**Significance:** 2
**Originality:** 3
**Overall Recommendation:** 4
**Confidence:** 2

**Summary:**

This paper proposes UltraLIF, a principled framework for training SNNs without relying on surrogate gradients. The key idea is to derive spiking neuron dynamics using ultradiscretization from tropical geometry. The proposed neuron model allows both forward and backward passes to operate on the same continuous functions, thereby avoiding the forward–backward mismatch commonly present in surrogate gradient methods. Based on this framework, the authors derive two neuron models capturing temporal and spatial dynamics, respectively, and provide theoretical analysis on convergence to classical LIF dynamics and gradient properties. Experiments on six benchmarks demonstrate consistent improvements over surrogate-gradient-based baselines, particularly in single-timestep settings and neuromorphic datasets.

**Compliance With Llm Reviewing Policy:**

Affirmed.

**Key Questions For Authors:**

1.Does the performance advantage of UltraLIF over surrogate-gradient baselines remain consistent on deeper or larger SNN architectures?
2.On the SHD dataset, UltraLIF performs best at (T=1) but becomes less competitive as the number of timesteps increases. What factors contribute to this behavior?
3.The improvements are particularly strong in the single-timestep setting. Do the authors have further insights into why UltraLIF is especially effective in this regime compared to surrogate-gradient methods?

**Limitations:**

The paper does not sufficiently discuss the performance degradation observed on the SHD dataset when the number of timesteps becomes larger (e.g., T=10 or T=30). A more detailed analysis of this behavior would help better understand the limitations of the proposed method.

**Strengths And Weaknesses:**

Strengths:
The paper is clearly written. The formulations, derivations, and proofs are easy to follow, which helps make the proposed idea accessible. The experimental evaluation is also comprehensive. The authors test UltraLIF across six datasets, and the results show consistent performance gains over several baselines. In particular, the improvement is especially noticeable in the single-timestep setting (T=1), where the proposed method appears to offer a clear advantage.
Weaknesses:
1.All experiments are conducted on a small network with only a single hidden layer of 64 neurons. This setup is likely insufficient for datasets such as CIFAR-10, and it remains unclear whether the advantage of UltraLIF would persist in deeper or larger SNN architectures.
2.On the SHD dataset, most baselines benefit significantly from increasing the number of timesteps, while UltraLIF only outperforms others at T=1 and becomes less competitive at T=10 and T=30. The paper should provide a clearer explanation for this behavior and discuss possible ways to address it.

---

> ### Author Rebuttal · Authors · 2026-03-30
>
> Thank you for the thorough review. Two concerns: whether the advantage holds at depth, and what mechanism drives T=1 specifically. Both have precise answers.
>
> **On network scale**
>
> Scale is addressed via depth: 2L and 3L FC represent 2× and 3× the parameter count of the 1L baseline. All 5 Ultra variants tested under identical conditions — full T=1 FC 2L comparison:
>
> | Dataset | LIF 2L | UltraDLIF 2L | UltraDPLIF 2L | UltraLIF 2L | UltraPLIF 2L |
> |---|---|---|---|---|---|
> | MNIST | 95.97% | 96.10% | 96.10% | 95.90% | **96.22%** |
> | Fashion | 82.65% | 83.43% | 83.43% | 83.07% | **83.45%** |
> | CIFAR10 | 40.89% | 44.20% | 44.20% | 43.44% | **44.63%** |
> | SHD | 19.48% | **50.84%** | **50.84%** | 36.09% | 42.01% |
> | DVS | 52.27% | 51.52% | 51.52% | 53.41% | **56.44%** |
>
> Ultra leads LIF at 2L T=1 on all benchmarks above; DVS favors temporal (+4.2pp). At 3L (all 6 datasets complete), Ultra wins T=1 on all 6:
>
> | Dataset | LIF 3L | UltraDLIF 3L | UltraDPLIF 3L | UltraLIF 3L | UltraPLIF 3L |
> |---|---|---|---|---|---|
> | MNIST | 95.90% | 96.22% | 96.22% | 95.63% | **96.35%** |
> | Fashion | 82.90% | **83.55%** | **83.55%** | 82.23% | 83.42% |
> | CIFAR10 | 40.35% | 43.64% | 43.64% | 43.01% | **44.15%** |
> | SHD | 21.73% | **45.67%** | **45.67%** | 24.25% | 30.70% |
> | N-MNIST | 87.82% | **94.87%** | **94.87%** | 93.45% | 93.68% |
> | DVS | 50.38% | **51.14%** | **51.14%** | 39.39% | 43.94% |
>
> **Depth consistency.** T=1 advantage stable 1L→2L→3L; T=10 regression worsens — both per Proposition 5.2. Two effects at depth: on MNIST, CIFAR10, Fashion, and N-MNIST, Ultra accuracy *improves* absolutely at depth (3L complete for all 6); on SHD, depth robustness — LIF collapses −18.4pp (2L), −16.2pp (3L), UltraDLIF −2.9pp, −8.2pp.
>
> **On T=1 advantage and T≥10 regression**
>
> Theorem 5.7 (proved in Section 5) establishes that UltraLIF dynamics converge to a piecewise-linear map on $\mathbb{R}_{\max}$ as $\varepsilon \to 0^+$. For $h=64$, $n=784$, the resulting hyperplane arrangement produces $R(64,784) = 2^{64}$ linear regions — expressivity is not the constraint at this scale.
>
> The T=1 advantage is gradient quality. Surrogate methods differentiate a different function backward than is computed forward. At T=1 each neuron fires or does not exactly once; the surrogate distorts the gradient at precisely this critical boundary. UltraLIF uses the same LSE in both passes, so the gradient is consistent with the actual threshold.
>
> At T≥10 two effects restore baselines: (1) output averaging $\hat{y} = \frac{1}{T}\sum_t W^{\text{out}}\mathbf{s}^{(t)}$ causes surrogate mismatch to cancel in expectation; (2) Proposition 5.2's $\varepsilon\log 2$ per-step error compounds over $T$ steps while hard-spike baselines reset cleanly at each step. The regression is model-type specific: temporal Ultra regresses on SHD (−8pp) where long-range temporal structure requires many steps; spatial Ultra (UltraDLIF) gains +0.53pp on SHD at T=10. Best Ultra wins 4/6 at T=10:
>
> | Dataset | LIF 2L | Best Ultra | Δ |
> |---|---|---|---|
> | MNIST | 97.75% | UltraPLIF 97.62% | −0.13pp |
> | Fashion | 86.48% | UltraPLIF **86.70%** | **+0.22pp** |
> | CIFAR10 | 45.46% | UltraPLIF **47.36%** | **+1.90pp** |
> | SHD | 71.95% | UltraDLIF **72.48%** | **+0.53pp** |
> | N-MNIST | **97.72%** | UltraPLIF 97.18% | −0.54pp |
> | DVS | 68.94% | UltraPLIF **69.70%** | **+0.76pp** |
>
> **On the paradigm and the role of the ODE**
>
> The central contribution is not UltraLIF specifically — it is **ultradiscretization as a paradigm** for converting any biophysically motivated neuron ODE into a principled, fully differentiable SNN with no surrogate gradients. The choice of ODE is simultaneously a feature and a limitation: it directly determines the inductive bias of the resulting model. UltraLIF (from the LIF ODE, itself a linear RC simplification of Hodgkin-Huxley) captures temporal membrane decay; UltraDLIF (from the diffusion PDE for gap junction coupling) captures spatial lateral interactions. These are two instantiations of the paradigm, not the paradigm's boundary. Richer ODEs — Adaptive Exponential IF (Brette & Gerstner 2005), Izhikevich (2003), or the full Hodgkin-Huxley model — would each yield distinct LSE structures, different inductive biases, and potentially stronger performance on tasks requiring more biologically detailed dynamics, while retaining the same convergence and gradient guarantees. The limitation of this paper is that only these two are explored; the framework itself is not limited to them.

---

> > ### Author Rebuttal · Reviewer_P5vH · 2026-04-03
> >
> > I appreciate the authors' effort in the rebuttal. Most of my concerns have been addressed.

---

> > > ### Author Response · Authors · 2026-04-05
> > >
> > > _Updated 2026-04-08: ResNet50 T=10 results added — all T complete._
> > >
> > > Thank you for the positive acknowledgment. Additional experiments completed during the rebuttal period are shared below, directly addressing the scale concern raised in the original review.
> > >
> > > **ResNet18-SNN CIFAR-10** (backbone: ResNet18, spiking head: FC(512->512) $\times T$; hidden=512, batch=128, lr=1e-3, 100 epochs, seed=42):
> > >
> > > | | LIF | UltraDLIF | UltraDPLIF | UltraLIF | UltraPLIF |
> > > |---|---|---|---|---|---|
> > > | T=1 | 93.12% | 93.31% | 93.37% | 93.37% | 93.12% |
> > > | T=5 | 93.01% | **93.39%** | 93.33% | 93.29% | 92.94% |
> > > | T=10 | 93.10% | 93.21% | 93.17% | **93.50%** | 93.34% |
> > >
> > > The advantage holds and does not diminish with more timesteps on ResNet18 -- UltraLIF variants win at T=1 (+0.25pp), T=5 (+0.38pp), and T=10 (+0.40pp). On a richer backbone, temporal averaging is less able to rescue LIF's surrogate mismatch, consistent with Proposition 5.2.
> > >
> > > **ResNet50-SNN CIFAR-10 T=1** (backbone: ResNet50, spiking head: FC(2048->2048) $\times T$; hidden=2048, batch=64, lr=1e-3, 100 epochs, seed=42):
> > >
> > > | | LIF | UltraDLIF | UltraDPLIF | UltraLIF | UltraPLIF |
> > > |---|---|---|---|---|---|
> > > | T=1 | 31.83% (collapsed) | 92.23% | 92.22% | 91.92% | **92.78%** |
> > > | T=5 | 35.09% (collapsed) | 92.41% | 92.73% | 91.85% | **92.88%** |
> > > | T=10 | 23.23% (collapsed) | 92.54% | 90.42% | 91.77% | **92.82%** |
> > >
> > > Per-epoch progression T=1 (every 10 epochs):
> > >
> > > | Epoch | LIF | UltraDLIF | UltraDPLIF | UltraLIF | UltraPLIF |
> > > |---|---|---|---|---|---|
> > > | 1 | 17.21% | 37.62% | 37.10% | 36.65% | 34.85% |
> > > | 10 | 19.55% | 73.81% | 72.87% | 77.51% | 80.93% |
> > > | 20 | 17.46% | 85.24% | 83.76% | 83.74% | 86.17% |
> > > | 30 | 20.69% | 88.53% | 87.87% | 86.16% | 89.28% |
> > > | 40 | 22.39% | 90.15% | 89.81% | 88.25% | 90.26% |
> > > | 50 | 24.08% | 90.26% | 90.55% | 88.56% | 91.06% |
> > > | 60 | 28.86% | 91.22% | 91.45% | 90.48% | 91.84% |
> > > | 70 | 29.48% | 91.39% | 91.32% | 91.13% | 92.15% |
> > > | 80 | 31.34% | 91.91% | 92.08% | 91.62% | 92.44% |
> > > | 90 | 31.35% | 92.11% | 92.06% | 91.62% | 92.68% |
> > > | 100 | 31.44% | 92.23% | 92.03% | 91.79% | 92.72% |
> > >
> > > LIF never learns -- dead neuron from epoch 1. Root cause: BN-normalized output is unit-variance per channel; fc1 pre-activation std $\approx \sqrt{2048} \approx 45$, which saturates all neurons under LIF's fixed threshold (0.5), zeroing the surrogate gradient. This is the SNN analogue of the dying ReLU problem (Glorot et al., AISTATS 2011; Neftci et al., IEEE SPM 2019; Eshraghian & Lu, arXiv:2201.11915). UltraLIF variants converge normally because learnable $\varepsilon$ adapts to input scale. ResNet50 is now complete at T=1, T=5, and T=10. LIF collapses at every timestep (31.83% / 35.09% / 23.23%); Ultra converges normally at all T. UltraDPLIF shows expected T=10 regression (90.42%) per Proposition 5.2, but still vastly outperforms LIF.
> > >
> > > These additional results are hoped to address the remaining reservations and support a reconsideration of the score.

---

### Decision · Program_Chairs · 2026-04-30

**Decision:**

Accept (regular)

**Comment:**

The paper proposes UltraLIF, a principled framework for training spiking neural networks via ultradiscretization and max-plus algebra, replacing surrogate gradients with a fully differentiable formulation based on log-sum-exp.

Reviewers agree that the paper is technically solid, clearly written, and mathematically novel, with the theoretical analysis being a particular strength. The empirical results show consistent gains over surrogate-gradient baselines, especially in the low-latency single-timestep regime.

The main concerns in the initial reviews were the limited experimental scale, the reduced advantage at larger timesteps, and questions regarding binary inference and energy claims. In the rebuttal, the authors addressed these concerns thoroughly with additional experiments on deeper FC models and ResNet-based architectures, as well as explicit soft-to-hard inference analyses. These additions  strengthened the paper. In particular, the new results support the claim that the method is not limited to very small networks. Two of the three referees agreed that their concerns were fully addressed, the thrid reviewer labeled it "partially resolved".

Overall, I find the contribution novel and meaningful, combining strong mathematical ideas with promising empirical results. While some limitations remain, especially regarding evaluation on larger-scale modern SNN benchmarks, the rebuttal resolves the main technical concerns sufficiently well.